# Distributed Low-rank Matrix Factorization
# With Exact Consensus

**Zhihui Zhu**\*
Mathematical Institute for Data Science
Johns Hopkins University
Baltimore, MD, USA
zzhu29@jhu.edu

**Qiuwei Li**\*
Department of Electrical Engineering
Colorado School of Mines
Golden, CO, USA
qiuli@mines.edu

**Xinshuo Yang**
Department of Electrical Engineering
Colorado School of Mines
Golden, CO, USA
xinshuoyang@mines.edu

**Gongguo Tang**
Department of Electrical Engineering
Colorado School of Mines
Golden, CO, USA
gtang@mines.edu

**Michael B. Wakin**
Department of Electrical Engineering
Colorado School of Mines
Golden, CO, USA
mwakin@mines.edu

## Abstract

Low-rank matrix factorization is a problem of broad importance, owing to the ubiquity of low-rank models in machine learning contexts. In spite of its non-convexity, this problem has a well-behaved geometric landscape, permitting local search algorithms such as gradient descent to converge to global minimizers. In this paper, we study low-rank matrix factorization in the distributed setting, where local variables at each node encode parts of the overall matrix factors, and consensus is encouraged among certain such variables. We identify conditions under which this new problem also has a well-behaved geometric landscape, and we propose an extension of distributed gradient descent (DGD) to solve this problem. The favorable landscape allows us to prove convergence to global optimality with exact consensus, a stronger result than what is provided by off-the-shelf DGD theory.

## 1 Introduction

A promising line of recent literature has examined the nonconvex objective functions that arise when certain matrix optimization problems are solved in factored form, that is, when a low-rank optimization variable $\mathbf{X}$ is replaced by a product of two thin matrices $\mathbf{U}\mathbf{V}^{\mathrm{T}}$ and the optimization proceeds jointly over $\mathbf{U}$ and $\mathbf{V}$ [6, 2, 10, 14, 23, 24, 27, 37, 38, 41, 42, 43]. In many cases, a study of the geometric landscape of these objective functions reveals that—despite their nonconvexity—they possess a certain favorable geometry. In particular, many of the resulting objective functions ($i$) satisfy the *strict saddle property* [13, 36], where every critical point is either a local minimum or a strict saddle point, at which the Hessian matrix has at least one negative eigenvalue, and ($ii$) have no spurious local minima (every local minimum corresponds to a global minimum).

One such problem—which is both of fundamental importance and representative of structures that arise in many other machine learning problems [19]—is the low-rank matrix approximation problem, where given a data matrix $\mathbf{Y}$ the objective is to minimize $\|\mathbf{U}\mathbf{V}^{\mathrm{T}} - \mathbf{Y}\|_F^2$. As we explain in Theorem 3.1, building on recent analysis in [32] and [42], this problem satisfies the strict saddle property and has no spurious local minima.

In parallel with the recent focus on the favorable geometry of certain nonconvex landscapes, it has been shown that a number of local search algorithms have the capability to avoid strict saddle points and converge to a local minimizer for problems that satisfy the strict saddle property [21, 17, 33, 25, 26, 28]; see [9, 11] for an overview. As stated in [20] and as we summarize in Theorem 2.2, gradient descent when started from a random initialization is one such algorithm. For problems such as low-rank matrix approximation that have no spurious local minima, converging to a local minimizer means converging to a global minimizer.

To date, the geometric and algorithmic research described above has largely focused on *centralized optimization*, where all computations happen at one "central" node that has full access, for example, to the data matrix $\mathbf{Y}$.

In this work, we study the impact of *distributing* the factored optimization problem, such as would be necessary if the data matrix $\mathbf{Y}$ in low-rank matrix approximation were partitioned into submatrices $\mathbf{Y} = [\mathbf{Y}_1 \quad \mathbf{Y}_2 \quad \cdots \quad \mathbf{Y}_J]$, each of which was available at only one node in a network. By similarly partitioning the matrix $\mathbf{V}$, one can partition the objective function

$$\|\mathbf{U}\mathbf{V}^{\mathrm{T}} - \mathbf{Y}\|_F^2 = \sum_{j=1}^{J} \|\mathbf{U}\mathbf{V}_j^{\mathrm{T}} - \mathbf{Y}_j\|_F^2. \tag{1}$$

One can attempt to minimize the resulting objective, in which the matrix $\mathbf{U}$ appears in every term of the summation, using techniques similar to classical distributed algorithms such as distributed gradient descent (DGD) [30], distributed Riemannian gradient descent (DRGD) [22], gossip-based methods [18, 4], and primal-dual methods [5, 35, 7, 15]. At a minimum, however, these algorithms involve creating local copies $\mathbf{U}^1, \mathbf{U}^2, \ldots, \mathbf{U}^J$ of the optimization variable $\mathbf{U}$ and iteratively sharing updates of these variables with the aim of converging to a consensus where (exactly or approximately) $\mathbf{U}^1 = \mathbf{U}^2 = \cdots = \mathbf{U}^J$. The introduction of additional variables (and possibly constraints) means that these distributed algorithms are navigating a potentially different geometric landscape than their centralized counterparts.

In this paper we study a straightforward extension of DGD for solving such problems. This extension, which we term DGD+LOCAL, resembles classical DGD in that each node $j$ has a local *copy* $\mathbf{U}^j$ of the optimization variable $\mathbf{U}$ as described above. Additionally, however, each node has a local *block* $\mathbf{V}_j$ of the partitioned optimization variable $\mathbf{V}$, and this block exists only locally at node $j$ without any consensus or sharing among other nodes. (In contrast, applying classical DGD to (1) would actually require every node to maintain and update a copy of both entire matrices $\mathbf{U}$ and $\mathbf{V}$.)

We present a geometric framework for analyzing the convergence of DGD+LOCAL in such problems. Our framework relies on a straightforward conversion which reveals (for example in the low-rank matrix approximation problem) that DGD+LOCAL as described above is equivalent to running conventional gradient descent on the objective function

$$\sum_{j=1}^{J} \left( \|\mathbf{U}^j\mathbf{V}_j^{\mathrm{T}} - \mathbf{Y}_j\|_F^2 + \sum_{i=1}^{J} w_{ji}\|\mathbf{U}^j - \mathbf{U}^i\|_F^2 \right), \tag{2}$$

where $w_{ji}$ are weights inherited from the DGD+LOCAL iterations. This objective function (2) differs from the original objective function (1) in two respects: it contains more optimization variables, and it includes a quadratic regularizer to encourage consensus. Although the geometry of (1) is understood to be well-behaved, new questions arise about the geometry of (2): Does it contain new critical points (local minima that are not global, saddle points that are not strict)? And on the consensus subspace, where $\mathbf{U}^1 = \mathbf{U}^2 = \cdots = \mathbf{U}^J$, how do the critical points of (2) relate to the critical points of (1)? We answer these questions and build on the algorithmic results for gradient descent to identify in Theorem 3.2 sufficient conditions where DGD+LOCAL is guaranteed to converge to a point that ($i$) is exactly on the consensus subspace, and ($ii$) coincides with a global minimizer of problem (1). Under these conditions, the distributed low-rank matrix approximation problem is shown to enjoy the same geometric and algorithmic guarantees as its well-behaved centralized counterpart.

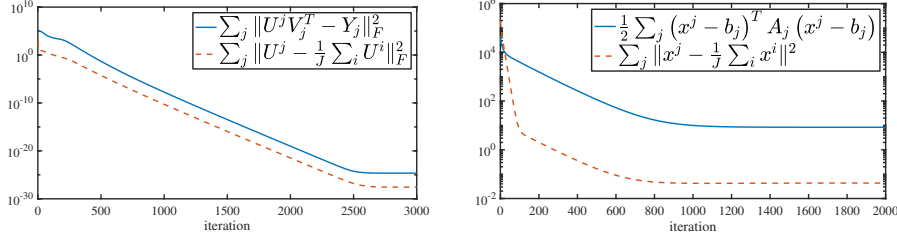

Figure 1: (Left) Exact optimality and consensus are possible for DGD+LOCAL on a distributed low-rank matrix approximation problem. (Right) Such properties do not hold in general, as demonstrated by running DGD on a least squares problem. Full details are provided in the supplementary material.

For the distributed low-rank matrix approximation problem, these guarantees are stronger than what appear in the literature for classical DGD and more general problems. In particular, we show exact convergence to the consensus subspace with a fixed DGD+LOCAL stepsize, which in more general works is accomplished only with diminishing DGD stepsizes for convex [8, 16] and nonconvex [40] problems or by otherwise modifying DGD as in the EXTRA algorithm [34]. Moreover, we show convergence to a global minimizer of the original centralized nonconvex problem. Until recently, existing DGD results either considered convex problems [8, 16] or showed convergence to stationary points of nonconvex problems [40]. Very recently, it was also shown [12] that with an appropriately small stepsize, DGD can converge to an arbitrarily small neighborhood of a second-order critical point for general nonconvex problems with additional technical assumptions. Our work differs from [12] in our use of DGD+LOCAL (rather than DGD) and our focus on one specific problem where we can establish stronger guarantees of exact global optimality and exact consensus without requiring an arbitrarily small (or diminishing) stepsize.

To summarize the above discussion, in this paper we make the following contributions:

- For general problems and under certain conditions (see Theorem 2.5 and Theorem 2.6), following the analysis in [21], we show that DGD+LOCAL converges to a second-order critical point of the regularized objective function (7). In the case of distributed low-rank matrix approximation, (7) corresponds to (2).

- For general problems and under a certain symmetric gradient condition (see Proposition 2.3), we show that every critical point of the distributed objective function (7) lies *exactly on* the consensus subspace, thus ensuring that DGD+LOCAL converges to an exact consensus point even with a fixed stepsize. We further show (see Theorem 2.7) that along the consensus subspace, the distributed objective function (7) has a certain geometric correspondence to its centralized counterpart (3). In particular, every critical point of (7) corresponds to a critical point of (3), and every strict saddle point of (3) corresponds to a strict saddle point of (7).

- We show (see Theorem 3.2) that distributed low-rank matrix approximation satisfies the symmetric gradient condition. Combined with the fact that the centralized low-rank matrix approximation problem satisfies the strict saddle property and has no spurious local minima (see Theorem 3.1), we conclude that DGD+LOCAL with a fixed stepsize achieves exact consensus and global optimality for distributed low-rank matrix approximation (see Theorem 3.2).

To demonstrate our conclusion for distributed low-rank matrix approximation, the left panel in Figure 1 shows the convergence of DGD+LOCAL for a low-rank matrix factorization problem whose setup is described in the supplementary material. Both the blue line (showing the objective value) and the red line (showing the consensus error) converge to zero. In contrast, the right panel in Figure 1 shows that DGD fails to achieve such optimality and consensus on a different, least squares problem. We also include experiments on distributed matrix completion and matrix sensing in the supplementary material.

Our main results on distributed low-rank matrix factorization are presented in Section 3. These results build on several more general algorithmic and geometric results that we first establish in Section 2. The results from Section 2 may have broader applicability, and the geometric and algorithmic discussions in Section 2 may have independent interest from one another. All proofs can be found in the supplementary material.

## 2 General Analysis of DGD+LOCAL

Consider a centralized minimization problem that can be written in the form

$$\underset{\mathbf{x}, \mathbf{y}}{\text{minimize}} \, f(\mathbf{x}, \mathbf{y}) = \sum_{j=1}^{J} f_j(\mathbf{x}, \mathbf{y}_j), \tag{3}$$

where $\mathbf{y} = \begin{bmatrix} \mathbf{y}_1^\mathsf{T} & \cdots & \mathbf{y}_J^\mathsf{T} \end{bmatrix}^\mathsf{T}$. Here $\mathbf{x}$ is the common variable in all of the objective functions $\{f_j\}_{j \in [J]}$ and $\mathbf{y}_j$ is the variable only corresponding to $f_j$.

The standard DGD algorithm [30] is stated for problems of minimizing $f(\mathbf{x}) = \sum_{j=1}^{J} f_j(\mathbf{x})$, and for such problems it involves updates of the form $\mathbf{x}^j(k+1) = \sum_{i=1}^{J} \left( \widetilde{w}_{ji} \mathbf{x}^i(k) \right) - \mu \nabla_{\mathbf{x}} f_j(\mathbf{x}^j(k))$, where $\{\widetilde{w}_{ji}\}$ are a set of symmetric nonnegative weights, and $\widetilde{w}_{ji}$ is positive if and only if nodes $i$ and $j$ are neighbors in the network or $i = j$. Throughout this paper, we will make the common assumption [29] that

$$\sum_{i=1}^{J} \widetilde{w}_{ji} = 1 \text{ for all } j \in [J]. \tag{4}$$

A very natural extension of DGD to problems of the form (3)—which involve local *copies* of the shared variable $\mathbf{x}$ and local *partitions* of the variable $\mathbf{y}$—is to perform the updates

$$\mathbf{x}^j(k+1) = \sum_{i=1}^{J} \left( \widetilde{w}_{ji} \mathbf{x}^i(k) \right) - \mu \nabla_{\mathbf{x}} f_j(\mathbf{x}^j(k), \mathbf{y}_j(k)),$$
$$\mathbf{y}_j(k+1) = \mathbf{y}_j(k) - \mu \nabla_{\mathbf{y}} f_j(\mathbf{x}^j(k), \mathbf{y}_j(k)). \tag{5}$$

Because we are interested in solving problems of the form (3), we refer to (5) as DGD+LOCAL throughout this paper. We note that DGD+LOCAL is not equivalent to algorithm would obtain by applying classical DGD to reach consensus over the concatenated variables $\mathbf{x}$ and $\mathbf{y}$ as this would require each node to maintain a local copy of the entire vector $\mathbf{y}$. For the same reason, DGD+LOCAL is not equivalent to the blocked variable problem described in [31].

### 2.1 Relation to Gradient Descent

Note that we can rewrite the first equation in (5) as

$$\mathbf{x}^j(k+1) = \mathbf{x}^j(k) - \mu \Big( \nabla_{\mathbf{x}} f_j(\mathbf{x}^j(k), \mathbf{y}_j(k)) + \sum_{i \neq j} \frac{\widetilde{w}_{ji}}{\mu} (\mathbf{x}^j(k) - \mathbf{x}^i(k)) \Big),$$

where the assumption (4) is used. Thus, by defining $\{w_{ji}\}$ such that

$$w_{ji} = w_{ij} = \begin{cases} \frac{\widetilde{w}_{ji}}{4\mu}, & i \neq j, \\ 0, & i = j, \end{cases} \tag{6}$$

we see that DGD+LOCAL (5) is equivalent to applying standard gradient descent (with stepsize $\mu$) to

$$\underset{\mathbf{z}}{\text{minimize}} \, g(\mathbf{z}) = \sum_{j=1}^{J} \Big( f_j(\mathbf{x}^j, \mathbf{y}_j) + \sum_{i=1}^{J} w_{ji} \|\mathbf{x}^j - \mathbf{x}^i\|_2^2 \Big), \tag{7}$$

where $\mathbf{z} = (\mathbf{x}^1, \dots, \mathbf{x}^J, \mathbf{y}_1, \dots, \mathbf{y}_J)$ and $\mathbf{W} = \{w_{ji}\}$ is a $J \times J$ connectivity matrix with nonnegative entries defined in (6) and zeros on the diagonal.

### 2.2 Algorithmic Analysis

We are interested in understanding the convergence of the gradient descent algorithm when it is applied to minimizing $g(\mathbf{z})$ in (7); as we have argued in Section 2.1, this is equivalent to running the DGD+LOCAL algorithm (5) to minimize the objective function $f(\mathbf{x}, \mathbf{y})$ in (3).

### 2.2.1 Objective Function Properties and Convergence of Gradient Descent

In this section, we review and present convergence results for gradient descent, before more explicitly discussing the convergence of DGD+LOCAL in Section 2.2.2. Under certain conditions, we can guarantee that gradient descent will converge to a second-order critical point of the objective function $g(\mathbf{z})$ in (7). The proof relies on certain properties of the functions $f_j$ comprising (3). We describe these properties before discussing convergence results. The first property concerns the assumption that each $f_j$ comprising (3) has Lipschitz gradient. In this case we can also argue that $g$ in (7) has Lipschitz gradient.

**Proposition 2.1.** *Let $f(\mathbf{x}, \mathbf{y}) = \sum_{j=1}^{J} f_j(\mathbf{x}, \mathbf{y}_j)$ be an objective function as in (3) and let $g(\mathbf{z})$ be as in (7) with $\mathbf{z} = (\mathbf{x}^1, \ldots, \mathbf{x}^J, \mathbf{y}_1, \ldots, \mathbf{y}_J)$. Suppose that each $f_j$ has Lipschitz gradient, i.e., $\nabla f_j$ is Lipschitz continuous with constant $L_j > 0$. Then $\nabla g$ is Lipschitz continuous with constant $L_g = L + \frac{2\omega}{\mu}$, where $L := \max_j L_j$, $\omega := \sum_{i \neq j}^{J} \widetilde{w}_{ji}$, and $\widetilde{w}_{ji}$ and $\mu$ are the DGD+LOCAL weights and stepsize as in (5).*

The second property concerns the following Łojasiewicz inequality.

**Definition 2.1.** *[1] Assume that $h : \mathbb{R}^n \to \mathbb{R}$ is continuously differentiable. Then $h$ is said to satisfy the Łojasiewicz inequality, if for any critical point $\overline{\mathbf{x}}$ of $h(\mathbf{x})$, there exist $\delta > 0$, $C_1 > 0$, and $\theta \in [0, 1)$ (which is often referred to as the KL exponent) such that*

$$|h(\mathbf{x}) - h(\overline{\mathbf{x}})|^{\theta} \leq C_1 \|\nabla h(\mathbf{x})\|, \ \ \forall \, \mathbf{x} \in B(\overline{\mathbf{x}}, \delta).$$

This Łojasiewicz inequality (or a more general Kurdyka-Łojasiewicz (KL) inequality for general nonsmooth problems) characterizes the local geometric properties of the objective function around its critical points and has proved useful for convergence analysis [1, 3]. The Łojasiewicz inequality (or KL inequality) is very general and holds for most problems in engineering. For example, every analytic function satisfies this Łojasiewicz inequality, but each function may have different Łojasiewicz exponent $\theta$ which determines the convergence rate; see [1, 3] for the details on this.

A general result for convergence of gradient descent to first-order critical point for a function satisfying the Łojasiewicz inequality is as follows.[2]

**Theorem 2.1.** *[1] Suppose $\inf_{\mathbb{R}^n} h > -\infty$ and $h$ satisfies the Łojasiewicz inequality. Also assume $\nabla h$ is Lipschitz continuous with constant $L > 0$. Let $\{\mathbf{x}(k)\}$ be the sequence generated by gradient descent $\mathbf{x}(k + 1) = \mathbf{x}(k) - \mu \nabla h(\mathbf{x}(k))$ with $\mu < \frac{1}{L}$. Then if the sequence $\{\mathbf{x}(k)\}$ is bounded, it converges to a critical point of $h$.*

The following result further characterizes the convergence behavior of gradient descent to a second-order critical point.

**Theorem 2.2.** *[21] Suppose $h$ is a twice-continuously differentiable function and $\nabla h$ is Lipschitz continuous with constant $L > 0$. Let $\{\mathbf{x}(k)\}$ be the sequence generated by gradient descent $\mathbf{x}(k + 1) = \mathbf{x}(k) - \mu \nabla h(\mathbf{x}(k))$ with $\mu < \frac{1}{L}$. Suppose $\mathbf{x}(0)$ is chosen randomly from a probability distribution supported on a set S having positive measure. Then the sequence $\{\mathbf{x}(k)\}$ almost surely avoids strict saddles, where the Hessian has at least one negative eigenvalue.*

Theorems 2.1 and 2.2 apply for functions $h$ that globally satisfy the Łojasiewicz and Lipschitz gradient conditions. In some problems, however, one or both of these properties may be satisfied only locally. Nevertheless, under an assumption of bounded iterations—as is already made in Theorem 2.1—it is possible to extend the first- and second-order convergence results to such functions. For example, one can extend Theorem 2.1 as follows by noting that the original derivation in [1] used the Łojasiewicz property only locally around limit points of the sequence $\{\mathbf{x}(k)\}$.

**Theorem 2.3.** *[1] Suppose $\inf_{\mathbb{R}^n} h > -\infty$. For $\rho > 0$, let $B_\rho$ denote the open ball of radius $\rho$: $B_\rho := \{\mathbf{x} : \|\mathbf{x}\|_2 < \rho\}$, and suppose $h$ satisfies the Łojasiewicz inequality at all points $\mathbf{x} \in B_\rho$. Also assume $\nabla h$ is Lipschitz continuous with constant $L > 0$. Let $\{\mathbf{x}(k)\}$ be the sequence generated by gradient descent $\mathbf{x}(k + 1) = \mathbf{x}(k) - \mu \nabla h(\mathbf{x}(k))$ with $\mu < \frac{1}{L}$. Suppose $\{\mathbf{x}(k)\} \subseteq B_\rho$ and all limit points of $\{\mathbf{x}(k)\}$ are in $B_\rho$. Then the sequence $\{\mathbf{x}(k)\}$ converges to a critical point of $h$.*

The following result establishes second-order convergence for a function with a locally Lipschitz gradient.

**Theorem 2.4.** *Let $\rho > 0$, and consider an objective function $h$ where:*

1. *$\inf_{\mathbb{R}^n} h > -\infty$,*

2. *$h$ satisfies the Łojasiewicz inequality within $B_\rho$,*

3. *$h$ is twice-continuously differentiable, and*

4. *$|h(\mathbf{x})| \le L_0$, $\|\nabla h(\mathbf{x})\| \le L_1$, and $\|\nabla^2 h(\mathbf{x})\|_2 \le L_2$ for all $\mathbf{x} \in B_{2\rho}$.*

*Suppose the gradient descent stepsize*

$$\mu < \frac{1}{L_2 + \frac{4L_1}{\rho} + \frac{(4+2\pi)L_0}{\rho^2}}. \tag{8}$$

*Suppose $\mathbf{x}(0)$ is chosen randomly from a probability distribution supported on a set $S \subseteq B_\rho$ with $S$ having positive measure, and suppose that under such random initialization, there is a positive probability that the sequence $\{\mathbf{x}(k)\}$ remains bounded in $B_\rho$ and all limit points of $\{\mathbf{x}(k)\}$ are in $B_\rho$. Then conditioned on observing that $\{\mathbf{x}(k)\} \subseteq B_\rho$ and all limit points of $\{\mathbf{x}(k)\}$ are in $B_\rho$, gradient descent converges to a critical point of $h$, and the probability that this critical point is a strict saddle point is zero.*

### 2.2.2 Convergence Analysis of DGD+LOCAL

As described in the following theorem, under certain conditions, we can guarantee that the DGD+LOCAL algorithm (5) (which is equivalent to gradient descent applied to minimizing $g(\mathbf{z})$ in (7)) will converge to a second-order critical point of the objective function $g(\mathbf{z})$.

**Theorem 2.5.** *Let $f(\mathbf{x}, \mathbf{y}) = \sum_{j=1}^{J} f_j(\mathbf{x}, \mathbf{y}_j)$ be an objective function as in (3) and let $g(\mathbf{z})$ be as in (7) with $\mathbf{z} = (\mathbf{x}^1, \dots, \mathbf{x}^J, \mathbf{y}_1, \dots, \mathbf{y}_J)$. Suppose each $f_j$ satisfies $\inf_{\mathbb{R}^n} f_j > -\infty$, is twice continuously-differentiable, and has Lipschitz gradient, i.e., $\nabla f_j$ is Lipschitz continuous with constant $L_j > 0$. Suppose $g$ satisfies the Łojasiewicz inequality. Let $L := \max_j L_j$, and let $\widetilde{w}_{ji}$ and $\mu$ be the DGD+LOCAL weights and stepsize as in (5). Assume $\omega := \max_j \sum_{i \ne j} \widetilde{w}_{ji} < \frac{1}{2}$. Let $\{\mathbf{z}(k)\}$ be the sequence generated by the DGD+LOCAL algorithm in (5) with*

$$\mu < \frac{1-2\omega}{L} \tag{9}$$

*and with random initialization from a probability distribution supported on a set $S$ having positive measure. Then if the sequence $\{\mathbf{z}(k)\}$ is bounded, it almost surely converges to a second-order critical point of the objective function in (7).*

*Remark* 2.1. The requirement that the DGD+LOCAL stepsize $\mu = O(\frac{1}{L})$ also appears in the convergence analysis of DGD in [39, 40].

*Remark* 2.2. The function $g$ is guaranteed to satisfy the Łojasiewicz inequality [1, 3], for example, if every $f_j$ is semi-algebraic, because this will imply that $g$ is semi-algebraic, and every semi-algebraic function satisfies the Łojasiewicz inequality.

*Remark* 2.3. The bounded sequence assumption is commonly used in analysis involving the Łojasiewicz inequality [1, 3]. If we further assume that the function is coercive, the sequence must be bounded.

*Remark* 2.4. Asymptotic convergence is a consequence of the Łojasiewicz inequality [1, 3]. Empirically, however, DGD+LOCAL for distributed low-rank matrix approximation converges at a linear rate; see Figure 1.

*Remark* 2.5. In order to satisfy (9), it must hold that $\omega < \frac{1}{2}$. In the case where the DGD+LOCAL weight matrix $\widetilde{\mathbf{W}}$ is symmetric and doubly stochastic (i.e., $\widetilde{\mathbf{W}}$ has nonnegative entries and each of its rows and columns sums to 1), this condition is equivalent to requiring that each diagonal element of $\widetilde{\mathbf{W}}$ is larger than $\frac{1}{2}$. Given any symmetric and doubly stochastic matrix $\widetilde{\mathbf{W}}$, one can design a new weight matrix $(\widetilde{\mathbf{W}} + \mathbf{I})/2$ that satisfies this requirement. This strategy is also mentioned at the end of [39, Section 2.1].

We also have the following DGD+LOCAL convergence result when the functions $f_j$ have only a locally Lipschitz gradient.

**Theorem 2.6.** *Let $f(\mathbf{x}, \mathbf{y}) = \sum_{j=1}^{J} f_j(\mathbf{x}, \mathbf{y}_j)$ be an objective function as in (3) and let $g(\mathbf{z})$ be as in (7) with $\mathbf{z} = (\mathbf{x}^1, \ldots, \mathbf{x}^J, \mathbf{y}_1, \ldots, \mathbf{y}_J)$. Let $\rho > 0$ and suppose each $f_j$ satisfies*

   1. $\inf_{\mathbb{R}^n} f_j > -\infty$,

   2. $f_j$ *is twice-continuously differentiable, and*

   3. $|f_j(\mathbf{x}, \mathbf{y}_j)| \leq L_{0,j}$, $\|\nabla f_j(\mathbf{x}, \mathbf{y}_j)\| \leq L_{1,j}$, *and* $\left\|\nabla^2 f_j(\mathbf{x}, \mathbf{y}_j)\right\|_2 \leq L_{2,j}$ *for all* $(\mathbf{x}, \mathbf{y}_j) \in B_{2\rho}$.

*Suppose also that $g$ satisfies the Łojasiewicz inequality within $B_\rho$. Let $\widetilde{w}_{ji}$ and $\mu$ be the DGD+LOCAL weights and stepsize as in (5). Assume $\omega := \max_j \sum_{i \neq j} \widetilde{w}_{ji} < \frac{1}{2}$. Let $\{\mathbf{z}(k)\}$ be the sequence generated by the DGD+LOCAL algorithm in (5) with*

$$\mu < \frac{1 - 2\omega}{\max_j L_{2,j} + \frac{4L_{1,j}}{\rho} + \frac{(4+2\pi)L_{0,j}}{\rho^2}}. \tag{10}$$

*Suppose $\mathbf{z}(0)$ is chosen randomly from a probability distribution supported on a set $S \subseteq B_\rho$ with $S$ having positive measure, and suppose that under such random initialization, there is a positive probability that the sequence $\{\mathbf{z}(k)\}$ remains bounded in $B_\rho$ and all limit points of $\{\mathbf{z}(k)\}$ are in $B_\rho$.*

*Then conditioned on observing that $\{\mathbf{z}(k)\} \subseteq B_\rho$ and all limit points of $\{\mathbf{z}(k)\}$ are in $B_\rho$, DGD+LOCAL converges to a critical point of the objective function in (7), and the probability that this critical point is a strict saddle point is zero.*

## 2.3 Geometric Analysis

Section 2.2 establishes that, under certain conditions, DGD+LOCAL will converge to a second-order critical point of the objective function $g(\mathbf{z})$ in (7).

In this section, we are interested in studying the geometric landscape of the distributed objective function in (7) and comparing it to the geometric landscape of the original centralized objective function in (3). In particular, we would like to understand how the critical points of $g(\mathbf{z})$ in (7) are related to the critical points of $f(\mathbf{x}, \mathbf{y})$ in (3).

These problems differ in two important respects:

- The objective function in (7) involves more optimization variables than that in (3). Thus, the optimization takes place in a higher-dimensional space and there is the potential for new features to be introduced into the geometric landscape.
- The objective function in (7) involves a quadratic regularization term that will promote consensus among the variables $\mathbf{x}^1, \ldots, \mathbf{x}^J$. This term is absent from (3). However, along the *consensus subspace* where $\mathbf{x}^1 = \cdots = \mathbf{x}^J$, this regularizer will be zero and the objective functions will coincide.

Despite these differences, we characterize below some ways in which the geometric landscapes of the two problems may be viewed as equivalent. These results may have independent interest from the specific DGD+LOCAL convergence analysis in Section 2.2.

Our first result establishes that if the sub-objective functions $f_j$ satisfy certain properties, the formulation (7) does not introduce any new global minima outside of the consensus subspace.

**Proposition 2.2.** *Let $f(\mathbf{x}, \mathbf{y}) = \sum_{j=1}^{J} f_j(\mathbf{x}, \mathbf{y}_j)$ be as in (3). Suppose the topology defined by $\mathbf{W}$ is connected. Also suppose there exist $\mathbf{x}^\star$ (which is independent of $j$) and $\mathbf{y}_j^\star, j \in [J]$ such that*

$$(\mathbf{x}^\star, \mathbf{y}_j^\star) \in \arg\min_{\mathbf{x}, \mathbf{y}_j} f_j(\mathbf{x}, \mathbf{y}_j), \ \forall \, j \in [J]. \tag{11}$$

*Then $g(\mathbf{z})$ defined in (7) satisfies $\min_{\mathbf{z}} g(\mathbf{z}) = \min_{\mathbf{x}, \mathbf{y}} f(\mathbf{x}, \mathbf{y})$, and $g(\mathbf{z})$ achieves its global minimum only for $\mathbf{z}$ with $\mathbf{x}^1 = \cdots = \mathbf{x}^J$.*

We note that the assumption in Proposition 2.2 is fairly strong, and while there are problems where it can hold, there are also many problems where it will not hold.

Proposition 2.2 establishes that, in certain cases, there will exist no global minimizers of the distributed objective function $g(\mathbf{z})$ that fall outside of the consensus subspace. (Moreover, and also importantly, there will *exist* a global minimizer *on* the consensus subspace.) Also relevant is the question of whether there may exist any *other* types of critical points (such as local minima or saddle points) outside of the consensus subspace. Under certain conditions, the following proposition ensures that the answer is no.

**Proposition 2.3.** *Let $f(\mathbf{x}, \mathbf{y})$ be as in (3) and $g(\mathbf{z})$ be as in (7) with $\mathbf{z} = (\mathbf{x}^1, \ldots, \mathbf{x}^J, \mathbf{y}_1, \ldots, \mathbf{y}_J)$. Suppose the matrix $\mathbf{W}$ is connected and symmetric. Also suppose the gradient of $f_j$ satisfies the following symmetric property:*

$$\langle \nabla_{\mathbf{x}} f_j(\mathbf{x}, \mathbf{y}_j), \mathbf{x} \rangle = \langle \nabla_{\mathbf{y}_j} f_j(\mathbf{x}, \mathbf{y}_j), \mathbf{y}_j \rangle \tag{12}$$

*for all $j \in [J]$. Then, any critical point of $g$ must satisfy $\mathbf{x}^1 = \cdots = \mathbf{x}^J$.*

Finally, we can also make a statement about the behavior of critical points that do fall on the consensus subspace.

**Theorem 2.7.** *Let $\mathcal{C}_f$ denote the set of critical points of (3): $\mathcal{C}_f := \{\mathbf{x}, \mathbf{y} : \nabla f(\mathbf{x}, \mathbf{y}) = \mathbf{0}\}$, and let $\mathcal{C}_g$ denote the set of critical points of (7): $\mathcal{C}_g := \left\{ \mathbf{z} : \nabla g(\mathbf{z}) = \mathbf{0} \right\}$. Then, for any $\mathbf{z} = (\mathbf{x}^1, \ldots, \mathbf{x}^J, \mathbf{y}) \in \mathcal{C}_g$ with $\mathbf{x}^1 = \cdots = \mathbf{x}^J = \mathbf{x}$, we have $(\mathbf{x}, \mathbf{y}) \in \mathcal{C}_f$. Furthermore, if $(\mathbf{x}, \mathbf{y})$ is a strict saddle of $f$, then $\mathbf{z} = (\mathbf{x}, \ldots, \mathbf{x}, \mathbf{y})$ is also a strict saddle of $g$.*

# 3 Analysis of Distributed Matrix Factorization

We now consider the prototypical low-rank matrix approximation in factored form, where given a data matrix $\mathbf{Y} \in \mathbb{R}^{n \times m}$, we seek to solve

$$\underset{\mathbf{U} \in \mathbb{R}^{n \times r}, \mathbf{V} \in \mathbb{R}^{m \times r}}{\text{minimize}} \|\mathbf{U}\mathbf{V}^{\mathrm{T}} - \mathbf{Y}\|_F^2. \tag{13}$$

Here $\mathbf{U} \in \mathbb{R}^{n \times r}$ and $\mathbf{V} \in \mathbb{R}^{m \times r}$ are tall matrices, and $r$ is chosen in advance to allow for a suitable approximation of $\mathbf{Y}$. In some of our results below, we will assume that the data matrix $\mathbf{Y}$ has rank at most $r$.

One can solve problem (13) using local search algorithms such as gradient descent. Such algorithms do not require expensive SVDs, and the storage complexity for $\mathbf{U}$ and $\mathbf{V}$ scales with $(n + m)r$, which is smaller than $nm$ as for $\mathbf{Y}$. Unfortunately, problem (13) is nonconvex in the optimization variables $(\mathbf{U}, \mathbf{V})$. Thus, the question arises of whether local search algorithms such as gradient descent actually converge to a global minimizer of (13). Using geometric analysis of the critical points of problem (13), however, it is possible to prove convergence to a global minimizer.

In Section 11 of the supplementary material, building on analysis in [32], we prove the following result about the favorable geometry of the nonconvex problem (13).

**Theorem 3.1.** *For any data matrix $\mathbf{Y}$, every critical point (i.e., every point where the gradient is zero) of problem (13) is either a global minimum or a strict saddle point, where the Hessian has at least one negative eigenvalue.*

Such favorable geometry has been used in the literature to show that local search algorithms (particularly gradient descent with random initialization [21]) will converge to a global minimum of the objective function.

## 3.1 Distributed Problem Formulation

We are interested in generalizing the matrix approximation problem from centralized to distributed scenarios. To be specific, suppose the columns of the data matrix $\mathbf{Y}$ are distributed among $J$ nodes/sensors. Without loss of generality, partition the columns of $\mathbf{Y}$ as

$$\mathbf{Y} = \begin{bmatrix} \mathbf{Y}_1 & \mathbf{Y}_2 & \cdots & \mathbf{Y}_J \end{bmatrix},$$

where for $j \in \{1, 2, \ldots, J\}$, matrix $\mathbf{Y}_j$ (which is stored at node $j$) has size $n \times m_j$, and where $m = \sum_{j=1}^{J} m_j$. Partitioning $\mathbf{V}$ similarly as

$$\mathbf{V} = \begin{bmatrix} \mathbf{V}_1^{\mathrm{T}} & \cdots & \mathbf{V}_J^{\mathrm{T}} \end{bmatrix}^{\mathrm{T}}, \tag{14}$$

where $\mathbf{V}_j$ has size $m_j \times r$, we obtain the following optimization problem

$$\underset{\mathbf{U}, \mathbf{V}_1, \ldots, \mathbf{V}_J}{\text{minimize}} \sum_{j=1}^{J} \|\mathbf{U}\mathbf{V}_j^{\mathrm{T}} - \mathbf{Y}_j\|_F^2, \tag{15}$$

which is exactly equivalent to (13). Problem (15), in turn, can be written in the form of problem (3) by taking

$$\mathbf{x} = \mathrm{vec}(\mathbf{U}), \quad \mathbf{y}_j = \mathrm{vec}(\mathbf{V}_j), \quad \text{and } f_j(\mathbf{x}, \mathbf{y}_j) = \|\mathbf{U}\mathbf{V}_j^{\mathrm{T}} - \mathbf{Y}_j\|_F^2. \tag{16}$$

Consequently, we can use the analysis from Section 2 to study the performance of DGD+LOCAL (5) when applied to problem (15).

For convenience, we note that in this context the DGD+LOCAL iterations (5) take the form

$$\mathbf{U}^j(k+1) = \sum_{i=1}^{J} \left( \widetilde{w}_{ji} \mathbf{U}^i(k) \right) - 2\mu (\mathbf{U}^j(k)\mathbf{V}_j^{\mathrm{T}}(k) - \mathbf{Y}_j)\mathbf{V}_j(k),$$

$$\mathbf{V}_j(k+1) = \mathbf{V}_j(k) - 2\mu (\mathbf{U}^j(k)\mathbf{V}_j^{\mathrm{T}}(k) - \mathbf{Y}_j)^{\mathrm{T}}\mathbf{U}^j(k), \tag{17}$$

and the corresponding gradient descent objective function (7) takes the form

$$\underset{\mathbf{z}}{\text{minimize}} \, g(\mathbf{z}) = \sum_{j=1}^{J} \left( \|\mathbf{U}^j\mathbf{V}_j^{\mathrm{T}} - \mathbf{Y}_j\|_F^2 + \sum_{i=1}^{J} w_{ji}\|\mathbf{U}^j - \mathbf{U}^i\|_F^2 \right), \tag{18}$$

where $\mathbf{U}^1, \ldots, \mathbf{U}^J \in \mathbb{R}^{n \times r}$ are local copies of the optimization variable $\mathbf{U}$; $\mathbf{V}_1, \ldots, \mathbf{V}_J$ are a partition of $\mathbf{V}$ as in (14); and the weights $\{w_{ji}\}$ are determined by $\{\widetilde{w}_{ji}\}$ and $\mu$ as in (6).

Problems (15) and (18) (as special cases of problems (3) and (7), respectively) satisfy many of the assumptions required for the geometric and algorithmic analysis in Section 2. We use these facts in proving our main result for the convergence of DGD+LOCAL on the matrix factorization problem.

**Theorem 3.2.** *Suppose rank$(\mathbf{Y}) \leq r$. Suppose DGD+LOCAL (17) is used to solve problem (15), with weights $\{\widetilde{w}_{ji}\}$ and stepsize*

$$\mu < \frac{1 - 2\omega}{\max_j \, (276 + 64\pi)\rho^2 + 34\|\mathbf{Y}_j\|_F + \frac{(8+4\pi)}{\rho^2}\|\mathbf{Y}_j\|_F^2} \tag{19}$$

*for some $\rho > 0$ and where $\omega := \max_j \sum_{i \neq j} \widetilde{w}_{ji} < \frac{1}{2}$. Suppose the $J \times J$ connectivity matrix $\mathbf{W} = \{w_{ji}\}$ (with $w_{ji}$ defined in (6)) is connected and symmetric. Let $\{\mathbf{z}(k)\}$ be the sequence generated by the DGD+LOCAL algorithm. Suppose $\mathbf{z}(0)$ is chosen randomly from a probability distribution supported on a set $S \subseteq B_\rho$ with $S$ having positive measure, and suppose that under such random initialization, there is a positive probability that the sequence $\{\mathbf{z}(k)\}$ remains bounded in $B_\rho$ and all limit points of $\{\mathbf{z}(k)\}$ are in $B_\rho$.*

*Then conditioned on observing that $\{\mathbf{z}(k)\} \subseteq B_\rho$ and all limit points of $\{\mathbf{z}(k)\}$ are in $B_\rho$, DGD+LOCAL almost surely converges to a solution $\mathbf{z}^\star = (\mathbf{U}^{1\star}, \ldots, \mathbf{U}^{J\star}, \mathbf{V}_1^\star, \ldots, \mathbf{V}_J^\star)$ with the following properties:*

- *Consensus: $\mathbf{U}^{1\star} = \cdots = \mathbf{U}^{J\star} = \mathbf{U}^\star$.*

- *Global optimality: $(\mathbf{U}^\star, \mathbf{V}^\star)$ is a global minimizer of (13), where $\mathbf{V}^\star$ denotes the concatenation of $\mathbf{V}_1^\star, \ldots, \mathbf{V}_J^\star$ as in (14).*

Such consensus and global optimality for the distributed low-rank matrix approximation problem are demonstrated by our experiment in the top panel of Figure 1.

**Acknowledgments**

This work was supported by the DARPA Lagrange Program under ONR/SPAWAR contract N660011824020. The views, opinions and/or findings expressed are those of the author(s) and should not be interpreted as representing the official views or policies of the Department of Defense or the U.S. Government.

## Footnotes

\*Equal contribution. ZZ is also with the Department of Electrical & Computer Engineering, University of Denver.

[2]The result in [1] is stated for the proximal method, but the result can be extended to gradient descent as long as $\mu < \frac{1}{L}$.

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
