[Supplementary Material]

# Supplementary Material

## 4    Proof of Proposition 2.1

Let $L = \max_j L_j$ and
$$\boldsymbol{\delta_z} = (\boldsymbol{\delta_{x^1}}, \ldots, \boldsymbol{\delta_{x^J}}, \boldsymbol{\delta_{y_1}}, \ldots, \boldsymbol{\delta_{y_J}}).$$
First, for any $\mathbf{z}$ and $\boldsymbol{\delta_z}$, and using the symmetry of $\mathbf{W} = \{w_{ij}\}$, we have
$$\nabla g(\mathbf{z} + \boldsymbol{\delta_z}) - \nabla g(\mathbf{z}) =$$

$$\begin{bmatrix} \nabla_{\mathbf{x}} f_1(\mathbf{x}^1 + \boldsymbol{\delta_{x^1}}, \mathbf{y}_1 + \boldsymbol{\delta_{y_1}}) - \nabla_{\mathbf{x}} f_1(\mathbf{x}^1, \mathbf{y}_1) + 4 \sum_{i=1}^{J} w_{1i}(\boldsymbol{\delta_{x^1}} - \boldsymbol{\delta_{x^i}}) \\ \vdots \\ \nabla_{\mathbf{x}} f_J(\mathbf{x}^J + \boldsymbol{\delta_{x^J}}, \mathbf{y}_J + \boldsymbol{\delta_{y_J}}) - \nabla_{\mathbf{x}} f_J(\mathbf{x}^J, \mathbf{y}_J) + 4 \sum_{i=1}^{J} w_{Ji}(\boldsymbol{\delta_{x^J}} - \boldsymbol{\delta_{x^i}}) \\ \nabla_{\mathbf{y}} f_1(\mathbf{x}^1 + \boldsymbol{\delta_{x^1}}, \mathbf{y}_1 + \boldsymbol{\delta_{y_1}}) - \nabla_{\mathbf{y}} f_1(\mathbf{x}^1, \mathbf{y}_1) \\ \vdots \\ \nabla_{\mathbf{y}} f_J(\mathbf{x}^J + \boldsymbol{\delta_{x^J}}, \mathbf{y}_J + \boldsymbol{\delta_{y_J}}) - \nabla_{\mathbf{y}} f_J(\mathbf{x}^J, \mathbf{y}_J) \end{bmatrix}$$

Then with some rearrangement, denoting $\nabla f_j = \nabla_{\left[\begin{smallmatrix}\mathbf{x}\\\mathbf{y}\end{smallmatrix}\right]} f_j$ and using the triangle inequality, we can obtain

$$\|\nabla g(\mathbf{z} + \boldsymbol{\delta_z}) - \nabla g(\mathbf{z})\|_2$$
$$\leq \left\| \begin{bmatrix} \nabla f_1(\mathbf{x}^1 + \boldsymbol{\delta_{x^1}}, \mathbf{y}_1 + \boldsymbol{\delta_{y_1}}) - \nabla f_1(\mathbf{x}^1, \mathbf{y}_1) \\ \vdots \\ \nabla f_J(\mathbf{x}^J + \boldsymbol{\delta_{x^J}}, \mathbf{y}_J + \boldsymbol{\delta_{y_J}}) - \nabla f_J(\mathbf{x}^J, \mathbf{y}_J) \end{bmatrix} \right\|_2$$
$$+ 4 \left\| \begin{bmatrix} \sum_{i=1}^{J} w_{1i}(\boldsymbol{\delta_{x^1}} - \boldsymbol{\delta_{x^i}}) \\ \vdots \\ \sum_{i=1}^{J} w_{Ji}(\boldsymbol{\delta_{x^J}} - \boldsymbol{\delta_{x^i}}) \end{bmatrix} \right\|_2$$
$$\leq \sqrt{\sum_{j=1}^{J} L_j^2 \left\| \begin{bmatrix} \boldsymbol{\delta_{x^j}} \\ \boldsymbol{\delta_{y_j}} \end{bmatrix} \right\|_2^2}$$
$$+ 4 \sqrt{\sum_{j=1}^{J} \left( \sum_{i=1}^{J} w_{ji} \right)^2 \|\boldsymbol{\delta_{x^j}}\|_2^2} + 4 \sqrt{\sum_{j=1}^{J} \left\| \sum_{i=1}^{J} w_{ji} \boldsymbol{\delta_{x^i}} \right\|_2^2}$$
$$\leq L\|\boldsymbol{\delta_z}\|_F + \left( 4 \max_j \sum_{i=1}^{J} w_{ji} \right) \left\| \begin{bmatrix} \boldsymbol{\delta_{x^1}} & \cdots & \boldsymbol{\delta_{x^1}} \end{bmatrix} \right\|_F$$
$$+ 4 \left( \max_j \sum_{i=1}^{J} w_{ji} \right) \left\| \begin{bmatrix} \boldsymbol{\delta_{x^1}} & \cdots & \boldsymbol{\delta_{x^1}} \end{bmatrix} \right\|_F .$$

where in the last inequality we use

$$\sqrt{\sum_{j=1}^{J} \left\| \sum_{i=j}^{J} w_{ji} \boldsymbol{\delta_{x^i}} \right\|_2^2} = \left\| \begin{bmatrix} \boldsymbol{\delta_{x^1}} & \cdots & \boldsymbol{\delta_{x^1}} \end{bmatrix} \mathbf{W} \right\|_F$$
$$= \left\| \mathbf{W}^{\mathrm{T}} \begin{bmatrix} \boldsymbol{\delta_{x^1}} & \cdots & \boldsymbol{\delta_{x^1}} \end{bmatrix}^{\mathrm{T}} \right\|_F \leq \|\mathbf{W}\| \left\| \begin{bmatrix} \boldsymbol{\delta_{x^1}} & \cdots & \boldsymbol{\delta_{x^1}} \end{bmatrix} \right\|_F$$
$$\leq \left( \max_j \sum_{i=1}^{J} w_{ji} \right) \left\| \begin{bmatrix} \boldsymbol{\delta_{x^1}} & \cdots & \boldsymbol{\delta_{x^1}} \end{bmatrix} \right\|_F$$

since $\|\mathbf{W}\| \leq \max_j \sum_{i \neq j} w_{ji} = \max_j \sum_{i=1}^{J} w_{ji}$ in view of that $\mathbf{W}$ is symmetric, $w_{ii} = 0$ and $w_{ij} \geq 0$ by (6).

Finally, using the definition of $w_{ji}$ (6), we have $\max_j \sum_{i=1}^{J} w_{ji} = \max_j \sum_{i \neq j}^{J} w_{ji} = \max_j \frac{\sum_{i \neq j}^{J} \widetilde{w}_{ji}}{4\mu} =: \frac{\omega}{4\mu}$, and further by the inequality $\left\| \begin{bmatrix} \boldsymbol{\delta_{x^1}} & \cdots & \boldsymbol{\delta_{x^1}} \end{bmatrix} \right\|_F \leq \|\boldsymbol{\delta_Z}\|_F$, we obtain that $\nabla g$ is Lipschitz continuous with constant

$$L_g = L + 4\left( \frac{\omega}{4\mu} \right) + 4\left( \frac{\omega}{4\mu} \right) = L + \frac{2\omega}{\mu}.$$

# 5 Proof of Theorem 2.4

The proof involves constructing a function $\widetilde{h}$ such that $\widetilde{h}(\mathbf{x}) = h(\mathbf{x})$ for all $\mathbf{x} \in B_\rho$ but where $\widetilde{h}$ has a globally Lipschitz gradient.

To do this, first define a window function $w : \mathbb{R}^n \to \mathbb{R}$,

$$
w\left(\mathbf{x}\right) = \begin{cases} 1, & \|\mathbf{x}\| \le \rho \\ 2 - \frac{\|\mathbf{x}\|}{\rho} + \frac{1}{2\pi}\sin\left(\frac{2\pi\|\mathbf{x}\|}{\rho}\right), & \rho < \|\mathbf{x}\| < 2\rho \\ 0, & \|\mathbf{x}\| \ge 2\rho, \end{cases}
$$

where $\|\cdot\| = \|\cdot\|_2$. Note that

$$
\nabla w\left(\mathbf{x}\right) = \begin{cases} 0, & \|\mathbf{x}\| \le \rho \\ -\frac{2\mathbf{x}}{\rho\|\mathbf{x}\|}\sin^2\left(\frac{\pi\|\mathbf{x}\|}{\rho}\right), & \rho < \|\mathbf{x}\| < 2\rho \\ 0, & \|\mathbf{x}\| \ge 2\rho \end{cases}
$$

and

$$
\nabla^2 w\left(\mathbf{x}\right)
$$

$$
= \begin{cases} 0, & \|\mathbf{x}\| \le \rho \\ -\frac{2}{\rho\|\mathbf{x}\|}\sin^2\left(\frac{\pi\|\mathbf{x}\|}{\rho}\right)I \\ \quad + \left(\frac{2}{\rho\|\mathbf{x}\|^3}\sin\left(\frac{\pi\|\mathbf{x}\|}{\rho}\right) - \frac{2\pi}{\rho^2\|\mathbf{x}\|^2}\sin\left(\frac{2\pi\|\mathbf{x}\|}{\rho}\right)\right)\mathbf{x}\mathbf{x}^{\mathrm{T}}, & \rho < \|\mathbf{x}\| < 2\rho \\ 0, & \|\mathbf{x}\| \ge 2\rho \end{cases}
$$

where $I$ denote the $n$-by-$n$ identity matrix. It is easy to verify that $w \in \mathcal{C}^2$ and $|w\left(\mathbf{x}\right)| \le 1$. To bound the gradient $\nabla w$, we have

$$
\|\nabla w\| = \left\|-\frac{2\mathbf{x}}{\rho\|\mathbf{x}\|}\sin^2\left(\frac{\pi\|\mathbf{x}\|}{\rho}\right)\right\| \le \frac{2}{\rho}.
$$

For the Hessian $\nabla^2 w$ with $\rho < \|\mathbf{x}\| < 2\rho$, we have

$$
\begin{aligned}
\|\nabla^2 w\| &\le \left\|\frac{2}{\rho\|\mathbf{x}\|}\sin^2\left(\frac{\pi\|\mathbf{x}\|}{\rho}\right)I\right\| \\
&\quad + \left\|\left(\frac{2}{\rho\|\mathbf{x}\|^3}\sin\left(\frac{\pi\|\mathbf{x}\|}{\rho}\right) - \frac{2\pi}{\rho^2\|\mathbf{x}\|^2}\sin\left(\frac{2\pi\|\mathbf{x}\|}{\rho}\right)\right)\mathbf{x}\mathbf{x}^{\mathrm{T}}\right\| \\
&\le \frac{4 + 2\pi}{\rho^2}.
\end{aligned}
$$

In fact, $\|\nabla^2 w\| \le \frac{4+2\pi}{\rho^2}$ for all $\mathbf{x}$ since $\nabla^2 w = 0$ for $\|\mathbf{x}\| \le \rho$ and $\|\mathbf{x}\| \ge 2\rho$. Now, we define $\widetilde{h}\left(\mathbf{x}\right) = h\left(\mathbf{x}\right)w\left(\mathbf{x}\right)$, which has the following properties:

- Since $h = \widetilde{h}$ in $B_\rho$, $\widetilde{h}$ satisfies the Łojasiewicz inequality in $B_\rho$.
- Since $h, w \in \mathcal{C}^2$, $\widetilde{h} \in \mathcal{C}^2$.
- Since $\inf_{\mathbb{R}^n} h > -\infty$ and $\inf_{\mathbb{R}^n} w > -\infty$, $\inf_{\mathbb{R}^n} \widetilde{h} > -\infty$.
- To globally bound the Lipschitz constant of the gradient of $\widetilde{h}$, note that

$$
\begin{aligned}
\left\|\nabla^2\widetilde{h}\right\| &= \left\|w\cdot\nabla^2 h + \nabla h\cdot(\nabla w)^{\mathrm{T}} + \nabla w\cdot(\nabla h)^{\mathrm{T}} + h\cdot\nabla^2 w\right\| \\
&\le |w|\left\|\nabla^2 h\right\| + 2\|\nabla w\|\|\nabla h\| + |h|\left\|\nabla^2 w\right\| \\
&\le L_2 + \frac{4L_1}{\rho} + \frac{(4+2\pi)L_0}{\rho^2}.
\end{aligned}
$$

Now consider the gradient descent algorithm with stepsize $\mu$ satisfying (8). Define

$$
\begin{aligned}
T_h = \{\mathbf{x}(0) \in B_\rho : \ &\text{all } \{\mathbf{x}(k)\} \subseteq B_\rho \text{ and all limit points of } \{\mathbf{x}(k)\} \\
&\text{are in } B_\rho \text{ when gradient descent is run on } h \text{ starting at } \mathbf{x}(0)\}
\end{aligned}
$$

and

$$T_{\widetilde{h}} = \{\mathbf{x}(0) \in B_\rho : \text{ all } \{\mathbf{x}(k)\} \subseteq B_\rho \text{ and all limit points of } \{\mathbf{x}(k)\}$$
$$\text{are in } B_\rho \text{ when gradient descent is run on } \widetilde{h} \text{ starting at } \mathbf{x}(0)\}.$$

Similarly, define

$$\Sigma_h = \{\mathbf{x}(0) \in B_\rho : \ \{\mathbf{x}(k)\} \text{ converges to a strict saddle}$$
$$\text{when gradient descent is run on } h \text{ starting at } \mathbf{x}(0)\}$$

and

$$\Sigma_{\widetilde{h}} = \{\mathbf{x}(0) \in B_\rho : \ \{\mathbf{x}(k)\} \text{ converges to a strict saddle}$$
$$\text{when gradient descent is run on } \widetilde{h} \text{ starting at } \mathbf{x}(0)\}.$$

Using the above properties, we see that Theorem 2.2 can be applied to $\widetilde{h}$, and so we conclude that $\Sigma_{\widetilde{h}}$ has measure zero.

Now, after running gradient descent on $h$ from a random initialization as in the theorem statement, condition on observing that $\{\mathbf{x}(k)\} \subseteq B_\rho$ and all limit points of $\{\mathbf{x}(k)\}$ are in $B_\rho$, i.e., that $\mathbf{x}(0) \in T_h$. Because $\{\mathbf{x}(k)\} \subseteq B_\rho$ and all limit points of $\{\mathbf{x}(k)\}$ are in $B_\rho$, and because $\{\mathbf{x}(k)\}$ matches the sequence that would be obtained by running gradient descent on $\widetilde{h}$, we can apply Theorem 2.3 to conclude that $\{\mathbf{x}(k)\}$ converges to a critical point of $\widetilde{h}$, and since this critical point belongs to $B_\rho$ and $\widetilde{h} = h$ inside $B_\rho$, we conclude that this is also a critical point of $h$.

Finally, using the definition of conditional probability, we have

$$P(\mathbf{x}(0) \in \Sigma_h | \mathbf{x}(0) \in T_h) = \frac{P(\mathbf{x}(0) \in \Sigma_h \cap T_h)}{P(\mathbf{x}(0) \in T_h)} = \frac{P(\mathbf{x}(0) \in \Sigma_{\widetilde{h}} \cap T_{\widetilde{h}})}{P(\mathbf{x}(0) \in T_h)},$$

where the second equality follows from the fact that $\widetilde{h} = h$ inside $B_\rho$: if a sequence of iterations stays bounded inside $B_\rho$ and converges to a strict saddle when gradient descent is run on $h$, the same will hold when gradient descent is run on $\widetilde{h}$, and vice versa. Since $\Sigma_{\widetilde{h}}$ has zero measure and because $\mathbf{x}(0)$ is chosen randomly from a probability distribution supported on a set $S \subseteq B_\rho$ with $S$ having positive measure, $P(\mathbf{x}(0) \in \Sigma_{\widetilde{h}} \cap T_{\widetilde{h}}) = 0$. Also, by assumption, $P(\mathbf{x}(0) \in T_h) > 0$. Therefore, $P(\mathbf{x}(0) \in \Sigma_h | \mathbf{x}(0) \in T_h) = \frac{0}{\text{nonzero}} = 0$.

# 6 Proof of Theorem 2.5

Recall that running the DGD+LOCAL algorithm (5) to minimize the objective function $f(\mathbf{x}, \mathbf{y})$ in (3) is equivalent to running gradient descent on $g(\mathbf{z})$ in (7). The proof is completed by invoking Theorem 2.1 and Theorem 2.2 with $h$ replaced by $g$. From Proposition 2.1, we have that $\nabla g$ is Lipschitz continuous with constant $L_g = L + \frac{2\omega}{\mu}$, and so choosing $\mu$ to satisfy (9) ensures that $\mu < \frac{1}{L_g}$ as required in Theorem 2.1 and Theorem 2.2.

# 7 Proof of Theorem 2.6

Recall that running the DGD+LOCAL algorithm (5) to minimize the objective function $f(\mathbf{x}, \mathbf{y})$ in (3) is equivalent to running gradient descent on $g(\mathbf{z})$ in (7). Similar to the approach taken in proving Theorem 2.4, to deal with the local Lipschitz condition, the proof involves constructing a function $\widetilde{g}$ such that $\widetilde{g}(\mathbf{z}) = g(\mathbf{z})$ for all $\mathbf{z} \in B_\rho$ but where $\widetilde{g}$ has a globally Lipschitz gradient.

To do this, recall the window function $w$ defined in Section 5 of the Supplementary material. Now, recall that

$$g(\mathbf{z}) = \sum_{j=1}^{J} \left( f_j(\mathbf{x}^j, \mathbf{y}_j) + \sum_{i=1}^{J} w_{ji} \|\mathbf{x}^j - \mathbf{x}^i\|_2^2 \right)$$

and define

$$\widetilde{g}(\mathbf{z}) = \sum_{j=1}^{J} \left( \widetilde{f}_j(\mathbf{x}^j, \mathbf{y}_j) + \sum_{i=1}^{J} w_{ji} \|\mathbf{x}^j - \mathbf{x}^i\|_2^2 \right), \tag{20}$$

where

$$\widetilde{f}_j(\mathbf{x}^j, \mathbf{y}_j) = f_j(\mathbf{x}^j, \mathbf{y}_j) w([(\mathbf{x}^j)^{\mathrm{T}} \quad \mathbf{y}_j^{\mathrm{T}}]^{\mathrm{T}}).$$

Since $\widetilde{f}_j(\mathbf{x}^j, \mathbf{y}_j) = f_j(\mathbf{x}^j, \mathbf{y}_j)$ for $(\mathbf{x}^j, \mathbf{y}_j) \in B_\rho$, we have that $\widetilde{g}(\mathbf{z}) = g(\mathbf{z})$ for all $\mathbf{z} \in B_\rho$.

We have the following properties for $\widetilde{g}$:

- Since $g = \widetilde{g}$ in $B_\rho$, $\widetilde{g}$ satisfies the Łojasiewicz inequality in $B_\rho$.
- Since $f_j \in C^2$ for all $j$ and $w \in C^2$, $\widetilde{g} \in C^2$.
- Since $\inf_{\mathbb{R}^n} f_j > -\infty$ for all $j$ and $\inf_{\mathbb{R}^n} w > -\infty$, $\inf_{\mathbb{R}^n} \widetilde{g} > -\infty$.
- To globally bound the Lipschitz constant of the gradient of $\widetilde{g}$, note that

$$
\begin{aligned}
\left\| \nabla^2 \widetilde{f}_j \right\| &= \left\| w \cdot \nabla^2 f_j + \nabla f_j \cdot (\nabla w)^{\mathrm{T}} + \nabla w \cdot (\nabla f_j)^{\mathrm{T}} + f_j \cdot \nabla^2 w \right\| \\
&\leq |w| \left\| \nabla^2 f_j \right\| + 2 \left\| \nabla w \right\| \left\| \nabla f_j \right\| + |f_j| \left\| \nabla^2 w \right\| \\
&\leq L_{2,j} + \frac{4 L_{1,j}}{\rho} + \frac{(4 + 2\pi) L_{0,j}}{\rho^2} \quad \text{for all } (\mathbf{x}^j, \mathbf{y}_j).
\end{aligned}
$$

Therefore, given the form of $\widetilde{g}$ in (20), we can conclude from Proposition 2.1 that globally, $\nabla \widetilde{g}$ is Lipschitz continuous with constant

$$
L_{\widetilde{g}} = \left( \max_j L_{2,j} + \frac{4 L_{1,j}}{\rho} + \frac{(4 + 2\pi) L_{0,j}}{\rho^2} \right) + \frac{2\omega}{\mu}.
$$

Now consider the gradient descent algorithm with stepsize $\mu$ satisfying (10). Define

$$
\begin{aligned}
T_g = \{ \mathbf{z}(0) \in B_\rho : \; &\text{all } \{\mathbf{z}(k)\} \subseteq B_\rho \text{ and all limit points of } \{\mathbf{z}(k)\} \\
&\text{are in } B_\rho \text{ when gradient descent is run on } g \text{ starting at } \mathbf{z}(0) \}
\end{aligned}
$$

and

$$
\begin{aligned}
T_{\widetilde{g}} = \{ \mathbf{z}(0) \in B_\rho : \; &\text{all } \{\mathbf{z}(k)\} \subseteq B_\rho \text{ and all limit points of } \{\mathbf{z}(k)\} \\
&\text{are in } B_\rho \text{ when gradient descent is run on } \widetilde{g} \text{ starting at } \mathbf{z}(0) \}.
\end{aligned}
$$

Similarly, define

$$
\begin{aligned}
\Sigma_g = \{ \mathbf{z}(0) \in B_\rho : \; &\{\mathbf{z}(k)\} \text{ converges to a strict saddle when} \\
&\text{gradient descent is run on } g \text{ starting at } \mathbf{z}(0) \}
\end{aligned}
$$

and

$$
\begin{aligned}
\Sigma_{\widetilde{g}} = \{ \mathbf{z}(0) \in B_\rho : \; &\{\mathbf{z}(k)\} \text{ converges to a strict saddle when} \\
&\text{gradient descent is run on } \widetilde{g} \text{ starting at } \mathbf{z}(0) \}.
\end{aligned}
$$

Using the above properties, we see that Theorem 2.2 can be applied to $\widetilde{g}$, and so we conclude that $\Sigma_{\widetilde{g}}$ has measure zero.

Now, after running gradient descent on $g$ from a random initialization as in the theorem statement, condition on observing that $\{\mathbf{z}(k)\} \subseteq B_\rho$ and all limit points of $\{\mathbf{z}(k)\}$ are in $B_\rho$, i.e., that $\mathbf{z}(0) \in T_g$. Because $\{\mathbf{z}(k)\} \subseteq B_\rho$ and all limit points of $\{\mathbf{z}(k)\}$ are in $B_\rho$, and because $\{\mathbf{z}(k)\}$ matches the sequence that would be obtained by running gradient descent on $\widetilde{g}$, we can apply Theorem 2.3 to conclude that $\{\mathbf{z}(k)\}$ converges to a critical point of $\widetilde{g}$, and since this critical point belongs to $B_\rho$ and $\widetilde{g} = g$ inside $B_\rho$, we conclude that this is also a critical point of $g$.

Finally, using the definition of conditional probability, we have

$$
\begin{aligned}
P(\mathbf{z}(0) \in \Sigma_g | \mathbf{z}(0) \in T_g) &= \frac{P(\mathbf{z}(0) \in \Sigma_g \cap T_g)}{P(\mathbf{z}(0) \in T_g)} \\
&= \frac{P(\mathbf{z}(0) \in \Sigma_{\widetilde{g}} \cap T_{\widetilde{g}})}{P(\mathbf{z}(0) \in T_g)},
\end{aligned}
$$

where the second equality follows from the fact that $\widetilde{g} = g$ inside $B_\rho$: if a sequence of iterations stays bounded inside $B_\rho$ and converges to a strict saddle when gradient descent is run on $g$, the same will hold when gradient descent is run on $\widetilde{g}$, and vice versa. Since $\Sigma_{\widetilde{g}}$ has zero measure and because $\mathbf{z}(0)$ is chosen randomly from a probability distribution supported on a set $S \subseteq B_\rho$ with $S$ having positive measure, $P(\mathbf{z}(0) \in \Sigma_{\widetilde{g}} \cap T_{\widetilde{g}}) = 0$. Also, by assumption, $P(\mathbf{z}(0) \in T_g) > 0$. Therefore, $P(\mathbf{z}(0) \in \Sigma_g | \mathbf{z}(0) \in T_g) = \frac{0}{\text{nonzero}} = 0$.

# 8 Proof of Proposition 2.2

First note that

$$\min_{\mathbf{z}} g(\mathbf{z}) = \sum_{j=1}^{J} \left( f_j(\mathbf{x}^j, \mathbf{y}_j) + \sum_{i=1}^{J} w_{ji} \|\mathbf{x}^j - \mathbf{x}^i\|_2^2 \right) \tag{21}$$

$$\geq \sum_{j=1}^{J} \min_{\mathbf{x}^j, \mathbf{y}_j} f_j(\mathbf{x}^j, \mathbf{y}_j) = \sum_{j=1}^{J} f_j(\mathbf{x}^\star, \mathbf{y}_j^\star) = \min_{\mathbf{x}, \mathbf{y}} f(\mathbf{x}, \mathbf{y}). \tag{22}$$

On the other hand, we have

$$\min_{\mathbf{z}} g(\mathbf{z}) = \min_{\mathbf{z}} \sum_{j=1}^{J} \left( f_j(\mathbf{x}^j, \mathbf{y}_j) + \sum_{i=1}^{J} w_{ji} \|\mathbf{x}^j - \mathbf{x}^i\|_2^2 \right)$$

$$\leq \min_{\mathbf{z}:\mathbf{x}^1 = \cdots = \mathbf{x}^J} \sum_{j=1}^{J} \left( f_j(\mathbf{x}^j, \mathbf{y}_j) + \sum_{i=1}^{J} w_{ji} \|\mathbf{x}^j - \mathbf{x}^i\|_2^2 \right)$$

$$= \min_{\mathbf{x}, \mathbf{y}} \sum_{j=1}^{J} f_j(\mathbf{x}, \mathbf{y}_j) = \min_{\mathbf{x}, \mathbf{y}} f(\mathbf{x}, \mathbf{y}).$$

Thus, we have

$$\min_{\mathbf{z}} g(\mathbf{z}) = \min_{\mathbf{x}, \mathbf{y}} f(\mathbf{x}, \mathbf{y}).$$

The proof is completed by noting that (22) achieves the equality only at $\mathbf{z}$ with $\mathbf{x}^1 = \cdots = \mathbf{x}^J$ since the topology defined by $\mathbf{W}$ is connected.

# 9 Proof of Proposition 2.3

The critical points of the objective function in (7) satisfy

$$\nabla_{\mathbf{x}^j} g(\mathbf{z}) = \nabla_{\mathbf{x}} f_j(\mathbf{x}^j, \mathbf{y}_j) + \sum_{i=1}^{J} 2 w_{ji}(\mathbf{x}^j - \mathbf{x}^i) = \mathbf{0}, \tag{23}$$

$$\nabla_{\mathbf{y}^j} g(\mathbf{z}) = \nabla_{\mathbf{y}_j} f_j(\mathbf{x}^j, \mathbf{y}_j) = \mathbf{0}, \forall \, j \in [J]. \tag{24}$$

Now taking the inner product of both sides in (23) with $\mathbf{x}^j$ and also the inner product of both sides in (24) with $\mathbf{y}^j$ and using the property (12), we have

$$\sum_{i=1}^{J} 2 w_{ji} \langle \mathbf{x}^j, \mathbf{x}^j - \mathbf{x}^i \rangle = 0$$

for all $j \in [J]$. Using the symmetric property of $\mathbf{W}$, we then have

$$\sum_{j=1}^{J} \sum_{i=1}^{J} w_{ji} \|\mathbf{x}^j - \mathbf{x}^i\|^2 = 0.$$

Therefore,

$$\mathbf{x}^i = \mathbf{x}^j, \text{ if } w_{ij} \neq 0$$

for any $i, j \in [J]$. Since the topology defined by $\mathbf{W}$ is connected, we finally have

$$\mathbf{x}^1 = \cdots = \mathbf{x}^J.$$

# 10 Proof of Theorem 2.7

We rewrite $\mathcal{C}_f$ as:

$$\mathcal{C}_f = \left\{ \mathbf{x}, \mathbf{y} : \sum_{j=1}^{J} \nabla_{\mathbf{x}} f_j(\mathbf{x}, \mathbf{y}_j) = \mathbf{0}, \nabla_{\mathbf{y}_j} f_j(\mathbf{x}, \mathbf{y}_j) = \mathbf{0}, \forall j \in [J] \right\}.$$

The critical points of the objective function in (7) satisfy

$$\nabla_{\mathbf{x}^j} g(\mathbf{z}) = \nabla_{\mathbf{x}} f_j(\mathbf{x}^j, \mathbf{y}_j) + \sum_{i=1}^{J} 2(w_{ij} + w_{ji})(\mathbf{x}^j - \mathbf{x}^i) = \mathbf{0},$$

$$\nabla_{\mathbf{y}_j} g(\mathbf{z}) = \nabla_{\mathbf{y}_j} f_j(\mathbf{x}^j, \mathbf{y}_j) = \mathbf{0}, \forall j \in [J].$$

With this, we rewrite $\mathcal{C}_g$ as

$$\mathcal{C}_g = \left\{ \mathbf{z} : \nabla_{\mathbf{x}} f_j(\mathbf{x}^j, \mathbf{y}_j) + \sum_{i=1}^{J} 2(w_{ij} + w_{ji})(\mathbf{x}^j - \mathbf{x}^i) = \mathbf{0}, \right.$$

$$\left. \nabla_{\mathbf{y}_j} f_j(\mathbf{x}^j, \mathbf{y}_j) = \mathbf{0}, \forall j \in [J] \right\}.$$

Thus, for any $\mathbf{z} = (\mathbf{x}^1, \dots, \mathbf{x}^J, \mathbf{y}) \in \mathcal{C}_g$ with $\mathbf{x}^1 = \cdots = \mathbf{x}^J = \mathbf{x}$, we have that $(\mathbf{x}, \mathbf{y})$ is a critical point of (3), i.e., $(\mathbf{x}, \mathbf{y}) \in \mathcal{C}_f$. In what follows, we check how the Hessian information (especially the smallest eigenvalue of the Hessian) of $(\mathbf{x}, \mathbf{y})$ is transformed to $\mathbf{z}$.

At any point $(\mathbf{x}, \mathbf{y})$, the Hessian quadratic form of $f$ for any $\mathbf{q_x}$ and $\mathbf{q_y} = \begin{bmatrix} \mathbf{q}_{\mathbf{y}_1}^{\mathrm{T}} & \cdots & \mathbf{q}_{\mathbf{y}_J}^{\mathrm{T}} \end{bmatrix}^{\mathrm{T}}$ is given by

$$[\nabla^2 f(\mathbf{x}, \mathbf{y})] \left( \begin{bmatrix} \mathbf{q_x} \\ \mathbf{q_y} \end{bmatrix}, \begin{bmatrix} \mathbf{q_x} \\ \mathbf{q_y} \end{bmatrix} \right) = \sum_{j=1}^{J} \nabla^2 f_j \left( \begin{bmatrix} \mathbf{q_x} \\ \mathbf{q}_{\mathbf{y}_j} \end{bmatrix}, \begin{bmatrix} \mathbf{q_x} \\ \mathbf{q}_{\mathbf{y}_j} \end{bmatrix} \right).$$

At any point $\mathbf{z}$, the Hessian quadratic form of $g$ for any

$$\mathbf{q} = \begin{bmatrix} \mathbf{q}_{\mathbf{x}^1}^{\mathrm{T}} & \cdots & \mathbf{q}_{\mathbf{x}^J}^{\mathrm{T}} & \mathbf{q}_{\mathbf{y}_1}^{\mathrm{T}} & \cdots & \mathbf{q}_{\mathbf{y}_J}^{\mathrm{T}} \end{bmatrix}$$

is given by

$$[\nabla^2 g(\mathbf{z})](\mathbf{q}, \mathbf{q}) = \sum_{j=1}^{J} \nabla^2 f_j \left( \begin{bmatrix} \mathbf{q}_{\mathbf{x}^j} \\ \mathbf{q}_{\mathbf{y}_j} \end{bmatrix}, \begin{bmatrix} \mathbf{q}_{\mathbf{x}^j} \\ \mathbf{q}_{\mathbf{y}_j} \end{bmatrix} \right) + \sum_{j=1}^{J} 2 w_{ji} \| \mathbf{q}_{\mathbf{x}^j} - \mathbf{q}_{\mathbf{x}^i} \|_2^2.$$

Now suppose $\lambda_{\min}(\nabla^2 f(\mathbf{x}, \mathbf{y})) < 0$ (where $\lambda_{\min}$ denotes the smallest eigenvalue), i.e., there exist $\mathbf{q_x}, \mathbf{q_y}$ such that

$$[\nabla^2 f(\mathbf{x}, \mathbf{y})] \left( \begin{bmatrix} \mathbf{q_x} \\ \mathbf{q_y} \end{bmatrix}, \begin{bmatrix} \mathbf{q_x} \\ \mathbf{q_y} \end{bmatrix} \right) < 0.$$

Choosing $\mathbf{q}_{\mathbf{x}^1} = \cdots = \mathbf{q}_{\mathbf{x}^J} = \mathbf{q_x}$, we have $[\nabla^2 g(\mathbf{z})](\mathbf{q}, \mathbf{q}) < 0$, i.e., $\lambda_{\min}(\nabla^2 g(\mathbf{z})) < 0$.

## 11 Proof of Theorem 3.1

Denote by $h(\mathbf{U}, \mathbf{V}) = \frac{1}{2} \|\mathbf{U}\mathbf{V}^{\mathrm{T}} - \mathbf{Y}\|_F^2$. Let $\mathcal{C}$ denote the set of critical points of $h$:

$$\mathcal{C} = \left\{ (\mathbf{U}, \mathbf{V}) : (\mathbf{U}\mathbf{V}^{\mathrm{T}} - \mathbf{Y})\mathbf{V} = \mathbf{0}, (\mathbf{U}\mathbf{V}^{\mathrm{T}} - \mathbf{Y})^{\mathrm{T}}\mathbf{U} = \mathbf{0} \right\}.$$

Our goal is to characterize the behavior of all the critical points that are not global minima. In particular, we want to show that every critical point of $h$ is either a global minimum or a strict saddle. Towards that end, we first recall the following result concerning the degenerate critical points.

**Lemma 11.1.** [32, Theorem 8 with $\mathbf{X} = \mathbf{I}$] *Any pair* $(\mathbf{U}, \mathbf{V}) \in \mathcal{C}$ *that is degenerate (i.e.,* $\mathrm{rank}(\mathbf{U}\mathbf{V}^{\mathrm{T}}) < r$*) is either a global minimum of $h$ (i.e.,* $\mathbf{U}\mathbf{V}^{\mathrm{T}} = \mathbf{Y}_r$ *where* $\mathbf{Y}_r$ *is a rank-r approximation of $\mathbf{Y}$) or a strict saddle (i.e.,* $\lambda_{\min}(\nabla^2 h(\mathbf{U}, \mathbf{V})) < 0$*).*

Note that the above result holds for any matrix $\mathbf{Y}$. When $\mathrm{rank}(\mathbf{Y}) \leq r$, then $\mathbf{Y}_r = \mathbf{Y}$. It follows from Lemma 11.1 that the behavior of all degenerate critical points is quite clear. For the remaining non-degenerate critical points, using the same argument in [42, Theorems 2–4], we first establish the following results concerning the critical points that are also balanced (i.e., $\mathbf{U}^{\mathrm{T}}\mathbf{U} = \mathbf{V}^{\mathrm{T}}\mathbf{V}$).

**Lemma 11.2.** [42, Theorems 2–4] *Any pair* $(\mathbf{U}, \mathbf{V}) \in \mathcal{C}$ *satisfying* $\mathbf{U}^{\mathrm{T}}\mathbf{U} = \mathbf{V}^{\mathrm{T}}\mathbf{V}$ *is either a global minimum of $h$ or a strict saddle.*

The above result also holds for any matrix $\mathbf{Y}$. With this result, we now show that non-degenerate critical points behave similarly to degenerate ones.

**Lemma 11.3.** *Any pair* $(\mathbf{U}, \mathbf{V}) \in \mathcal{C}$ *that is non-degenerate (i.e.,* $\mathrm{rank}(\mathbf{U}\mathbf{V}^{\mathrm{T}}) = r$*) is either a global minimum of $h$ or a strict saddle.*

*Proof of Lemma 11.3.* Suppose $(\mathbf{U}, \mathbf{V})$ is not a global minimum of $h$. Let $\mathbf{U}\mathbf{V}^{\mathrm{T}} = \mathbf{P}\mathbf{\Sigma}\mathbf{Q}^{\mathrm{T}}$ be a reduced SVD of $\mathbf{U}\mathbf{V}^{\mathrm{T}}$. Since $\mathrm{rank}(\mathbf{U}\mathbf{V}^{\mathrm{T}}) = r$ and both $\mathbf{U}$ and $\mathbf{V}$ have only $r$ columns, we know $\mathrm{rank}(\mathbf{U}) = \mathrm{rank}(\mathbf{V}) = r$. Denote by $\mathbf{D} = (\mathbf{U}^{\mathrm{T}}\mathbf{U})^{-1}\mathbf{U}^{\mathrm{T}}\mathbf{P}\mathbf{\Sigma}^{1/2}$ and $\mathbf{G} = (\mathbf{V}^{\mathrm{T}}\mathbf{V})^{-1}\mathbf{V}^{\mathrm{T}}\mathbf{Q}\mathbf{\Sigma}^{1/2}$. With this, we have

$$\mathbf{D}\mathbf{G}^{\mathrm{T}} = (\mathbf{U}^{\mathrm{T}}\mathbf{U})^{-1}\mathbf{U}^{\mathrm{T}}\mathbf{P}\mathbf{\Sigma}\mathbf{Q}^{\mathrm{T}}\mathbf{V}(\mathbf{V}^{\mathrm{T}}\mathbf{V})^{-1} = \mathbf{I},$$

and

$$\widetilde{\mathbf{U}} = \mathbf{U}\mathbf{D} = \mathbf{P}\mathbf{\Sigma}^{1/2}, \ \widetilde{\mathbf{V}} = \mathbf{V}\mathbf{G} = \mathbf{Q}\mathbf{\Sigma}^{1/2}.$$

The above constructed pair $(\widetilde{\mathbf{U}}, \widetilde{\mathbf{V}})$ satisfies

$$\widetilde{\mathbf{U}}\widetilde{\mathbf{V}}^{\mathrm{T}} = \mathbf{U}\mathbf{V}^{\mathrm{T}}, \ \widetilde{\mathbf{U}}^{\mathrm{T}}\widetilde{\mathbf{U}} = \widetilde{\mathbf{V}}^{\mathrm{T}}\widetilde{\mathbf{V}}.$$

Since $(\mathbf{U}, \mathbf{V}) \in \mathcal{C}$, we have

$$\nabla h_{\mathbf{U}}(\widetilde{\mathbf{U}}, \widetilde{\mathbf{V}}) = \nabla h_{\mathbf{U}}(\mathbf{U}, \mathbf{V})\mathbf{D} = \mathbf{0}, \ \nabla h_{\mathbf{V}}(\widetilde{\mathbf{U}}, \widetilde{\mathbf{V}}) = \nabla h_{\mathbf{V}}(\mathbf{U}, \mathbf{V})\mathbf{G} = \mathbf{0},$$

which implies that $(\widetilde{\mathbf{U}}, \widetilde{\mathbf{V}})$ is also a critical point (but not a global minimum since by assumption $(\mathbf{U}, \mathbf{V})$ is not a global minimum) of $h$. Since $(\widetilde{\mathbf{U}}, \widetilde{\mathbf{V}})$ is also balanced, it follows from Lemma 11.2 that there exists $\widetilde{\mathbf{\Delta}}_{\widetilde{\mathbf{U}}}$ and $\widetilde{\mathbf{\Delta}}_{\widetilde{\mathbf{V}}}$ such that

$$[\nabla^2 h(\widetilde{\mathbf{U}}, \widetilde{\mathbf{V}})](\widetilde{\mathbf{\Delta}}, \widetilde{\mathbf{\Delta}}) < 0.$$

Now construct $\mathbf{\Delta}_{\mathbf{U}} = \widetilde{\mathbf{\Delta}}_{\widetilde{\mathbf{U}}}\mathbf{D}^{-1}$ and $\mathbf{\Delta}_{\mathbf{V}} = \widetilde{\mathbf{\Delta}}_{\widetilde{\mathbf{V}}}\mathbf{G}^{-1}$. Then, we have

$$\begin{aligned}
[\nabla^2 h(\mathbf{U}, \mathbf{V})](\mathbf{\Delta}, \mathbf{\Delta}) &= \|\mathbf{\Delta}_{\mathbf{U}}\mathbf{V}^{\mathrm{T}} + \mathbf{U}\mathbf{\Delta}_{\mathbf{V}}^{\mathrm{T}}\|_F^2 + 2\langle \mathbf{U}\mathbf{V}^{\mathrm{T}} - \mathbf{Y}, \mathbf{\Delta}_{\mathbf{U}}\mathbf{\Delta}_{\mathbf{V}}^{\mathrm{T}}\rangle \\
&= \|\widetilde{\mathbf{\Delta}}_{\widetilde{\mathbf{U}}}\widetilde{\mathbf{V}}^{\mathrm{T}} + \widetilde{\mathbf{U}}\widetilde{\mathbf{\Delta}}_{\widetilde{\mathbf{V}}}^{\mathrm{T}}\|_F^2 + 2\langle \widetilde{\mathbf{U}}\widetilde{\mathbf{V}}^{\mathrm{T}} - \mathbf{Y}, \widetilde{\mathbf{\Delta}}_{\widetilde{\mathbf{U}}}\widetilde{\mathbf{\Delta}}_{\widetilde{\mathbf{V}}}^{\mathrm{T}}\rangle \\
&= [\nabla^2 h(\widetilde{\mathbf{U}}, \widetilde{\mathbf{V}})](\widetilde{\mathbf{\Delta}}, \widetilde{\mathbf{\Delta}}) < 0,
\end{aligned}$$

which implies that $(\mathbf{U}, \mathbf{V})$ is a strict saddle. $\qquad\square$

Lemma 11.2 together with Lemma 11.3 implies that any pair $(\mathbf{U}, \mathbf{V}) \in \mathcal{C}$ is either a global minimum of $h$ or a strict saddle.

# 12 Proof of Theorem 3.2

We begin by arguing that DGD+LOCAL converges almost surely (when $\mathbf{z}(0)$ is chosen randomly inside $B_\rho$) to a second-order critical point of (18). To do this, our goal is to invoke Theorem 2.6. We note that each $f_j$ defined in (16) satisfies $\inf_{\mathbf{U}, \mathbf{V}_j} f_j > -\infty$ and is twice-continuously differentiable. Also, since the functions $f_j$ are semi-algebraic, $g$ satisfies the Łojasiewicz inequality globally. The functions $f_j$ do not have globally Lipschitz gradient. However, we can find quantities $L_{0,j}, L_{1,j}, L_{2,j}$ such that $|f_j(\mathbf{x}, \mathbf{y}_j)| \le L_{0,j}$, $\|\nabla f_j(\mathbf{x}, \mathbf{y}_j)\| \le L_{1,j}$, and $\|\nabla^2 f_j(\mathbf{x}, \mathbf{y})\|_2 \le L_{2,j}$ for all $(\mathbf{x}, \mathbf{y}_j) \in B_{2\rho}$. For $L_{0,j}$:

$$\begin{aligned}
|f_j(\mathbf{x}, \mathbf{y}_j)| &= \|\mathbf{U}\mathbf{V}_j^{\mathrm{T}} - \mathbf{Y}_j\|_F^2 \\
&\le (\|\mathbf{U}\mathbf{V}_j^{\mathrm{T}}\|_F + \|\mathbf{Y}_j\|_F)^2 \\
&\le (\|\mathbf{U}\|_F \|\mathbf{V}^j\|_F + \|\mathbf{Y}_j\|_F)^2 \\
&\le (4\rho^2 + \|\mathbf{Y}_j\|_F)^2 \\
&\le 32\rho^4 + 2\|\mathbf{Y}_j\|_F^2.
\end{aligned}$$

For $L_{1,j}$:

$$\begin{aligned}
\|\nabla f_j(\mathbf{x}, \mathbf{y}_j)\| &= \left\| \begin{bmatrix} \nabla_{\mathbf{U}} \|\mathbf{U}\mathbf{V}_j^{\mathrm{T}} - \mathbf{Y}_j\|_F^2 \\ \nabla_{\mathbf{V}_j} \|\mathbf{U}\mathbf{V}_j^{\mathrm{T}} - \mathbf{Y}_j\|_F^2 \end{bmatrix} \right\|_F \\
&= \left\| \begin{bmatrix} 2(\mathbf{U}\mathbf{V}_j^{\mathrm{T}} - \mathbf{Y}_j)\mathbf{V}_j \\ 2(\mathbf{U}\mathbf{V}_j^{\mathrm{T}} - \mathbf{Y}_j)^{\mathrm{T}}\mathbf{U} \end{bmatrix} \right\|_F \\
&\le 2\left(\|\mathbf{U}\mathbf{V}_j^{\mathrm{T}}\mathbf{V}_j\|_F + \|\mathbf{Y}_j\mathbf{V}_j\|_F + \|\mathbf{V}_j\mathbf{U}^{\mathrm{T}}\mathbf{U}\|_F + \|\mathbf{Y}_j^{\mathrm{T}}\mathbf{U}\|_F\right) \\
&\le 2\left(8\rho^3 + 2\rho\|\mathbf{Y}_j\|_F + 8\rho^3 + 2\rho\|\mathbf{Y}_j\|_F\right) \\
&= 32\rho^3 + 8\rho\|\mathbf{Y}_j\|_F.
\end{aligned}$$

For $L_{2,j}$, we can bound the Lipschitz constant of $\nabla f_j$ in $B_{2\rho}$ as follows. Denote $\mathbf{D} = \begin{bmatrix} \mathbf{D_U} \\ \mathbf{D_{V_j}} \end{bmatrix}$. Then

$$\frac{1}{2}\|\nabla^2 f_j(\mathbf{U}, \mathbf{V}_j)\| = \frac{1}{2} \max_{\|\mathbf{D}\|_F=1} [\nabla^2 f_j(\mathbf{U}, \mathbf{V}_j)](\mathbf{D}, \mathbf{D})$$

$$= \max_{\|\mathbf{D}\|_F=1} \|\mathbf{D_U}\mathbf{V}_j^{\mathrm{T}} + \mathbf{U}\mathbf{D}_{\mathbf{V}_j}^{\mathrm{T}}\|_F^2 + 2\langle \mathbf{U}\mathbf{V}_j^{\mathrm{T}}, \mathbf{D_U}\mathbf{D}_{\mathbf{V}_j}^{\mathrm{T}}\rangle - 2\langle \mathbf{Y}_j, \mathbf{D_U}\mathbf{D}_{\mathbf{V}_j}^{\mathrm{T}}\rangle$$

$$\leq \max_{\|\mathbf{D}\|_F=1} \Bigg( \frac{5}{2}(\|\mathbf{V}_j\|_F^2 + \|\mathbf{U}\|_F^2)(\|\mathbf{D_U}\|_F^2$$

$$+ \|\mathbf{D}_{\mathbf{V}_j}\|_F^2) + \|\mathbf{Y}_j\|_F(\|\mathbf{D_U}\|_F^2 + \|\mathbf{D}_{\mathbf{V}_j}\|_F^2) \Bigg)$$

$$\leq \max_{\|\mathbf{D}\|_F=1} (10\rho^2 + \|\mathbf{Y}_j\|_F)(\|\mathbf{D_U}\|_F^2 + \|\mathbf{D}_{\mathbf{V}_j}\|_F^2) = 10\rho^2 + \|\mathbf{Y}_j\|_F,$$

where the last inequality holds because $\|\mathbf{U}\|_F^2 + \|\mathbf{V}_j\|_F^2 \leq 4\rho^2$. Therefore we can bound the Lipschitz constant of $\nabla f_j$ as $L_j \leq 20\rho^2 + 2\|\mathbf{Y}_j\|_F$ for all $(\mathbf{U}, \mathbf{V}_j)$ such that $\|\mathbf{U}\|_F^2 + \|\mathbf{V}_j\|_F^2 \leq 4\rho^2$. Now,

$$L_{2,j} + \frac{4L_{1,j}}{\rho} + \frac{(4 + 2\pi) L_{0,j}}{\rho^2}$$

$$= 20\rho^2 + 2\|\mathbf{Y}_j\|_F + \frac{4}{\rho}(32\rho^3 + 8\rho\|\mathbf{Y}_j\|_F) + \frac{(4 + 2\pi)}{\rho^2}(32\rho^4 + 2\|\mathbf{Y}_j\|_F^2)$$

$$= 20\rho^2 + 2\|\mathbf{Y}_j\|_F + 128\rho^2 + 32\|\mathbf{Y}_j\|_F + (128 + 64\pi)\rho^2 + \frac{(8 + 4\pi)}{\rho^2}\|\mathbf{Y}_j\|_F^2$$

$$= (276 + 64\pi)\rho^2 + 34\|\mathbf{Y}_j\|_F + \frac{(8 + 4\pi)}{\rho^2}\|\mathbf{Y}_j\|_F^2.$$

Thus, choosing $\mu$ to satisfy (19) ensures that (10) is met.

From Theorem 2.6, we then conclude that conditioned on observing that $\{\mathbf{z}(k)\} \subseteq B_\rho$ and all limit points of $\{\mathbf{z}(k)\}$ are in $B_\rho$, DGD+LOCAL converges to a critical point of the objective function in (18), and the probability that this critical point is a strict saddle point is zero. We refer to this point as $\mathbf{z}^\star$.

Next, note that the assumption of Proposition 2.2 is satisfied if $\mathbf{Y}$ has rank at most $r$. In particular, there exist $\widetilde{\mathbf{U}}, \widetilde{\mathbf{V}}$ such that $\widetilde{\mathbf{U}}\widetilde{\mathbf{V}}^{\mathrm{T}} = \mathbf{Y}$ and so we may take $\mathbf{x}^\star = \text{vec}(\widetilde{\mathbf{U}})$ and $\mathbf{y}_j^\star = \text{vec}(\widetilde{\mathbf{V}}_j)$ to achieve $f_j(\mathbf{x}^\star, \mathbf{y}_j^\star) = 0$, which is the smallest possible value for each $f_j$. Proposition 2.2 thus guarantees that (18) has at least one critical point that is not a strict saddle (and in fact that it is a global minimizer that falls on the consensus subspace).

Next, note that the symmetric property required for Proposition 2.3 is satisfied. To see this, observe that

$$\nabla_{\mathbf{U}}\|\mathbf{U}\mathbf{V}_j^{\mathrm{T}} - \mathbf{Y}_j\|_F^2 = 2(\mathbf{U}\mathbf{V}_j^{\mathrm{T}} - \mathbf{Y}_j)\mathbf{V}_j$$

and

$$\nabla_{\mathbf{V}_j}\|\mathbf{U}\mathbf{V}_j^{\mathrm{T}} - \mathbf{Y}_j\|_F^2 = 2(\mathbf{U}\mathbf{V}_j^{\mathrm{T}} - \mathbf{Y}_j)^{\mathrm{T}}\mathbf{U}.$$

Thus,

$$\langle \nabla_{\mathbf{U}}\|\mathbf{U}\mathbf{V}_j^{\mathrm{T}} - \mathbf{Y}_j\|_F^2, \mathbf{U}\rangle = 2 \cdot \text{tr}(U^{\mathrm{T}}(\mathbf{U}\mathbf{V}_j^{\mathrm{T}} - \mathbf{Y}_j)\mathbf{V}_j)$$

$$= 2 \cdot \text{tr}(\mathbf{V}_j^{\mathrm{T}}(\mathbf{U}\mathbf{V}_j^{\mathrm{T}} - \mathbf{Y}_j)^{\mathrm{T}}\mathbf{U}) = \langle \nabla_{\mathbf{V}_j}\|\mathbf{U}\mathbf{V}_j^{\mathrm{T}} - \mathbf{Y}_j\|_F^2, \mathbf{V}_j\rangle.$$

Proposition 2.3 thus guarantees that (18) has no critical points outside of the consensus subspace. Since we have argued that DGD+LOCAL converges to a second-order critical point $\mathbf{z}^\star$ of (18), it follows that $\mathbf{z}^\star$ must be on the consensus subspace; that is, $\mathbf{z}^\star = (\mathbf{U}^{1\star}, \ldots, \mathbf{U}^{J\star}, \mathbf{V}_1^\star, \ldots, \mathbf{V}_J^\star)$ with $\mathbf{U}^{1\star} = \cdots = \mathbf{U}^{J\star} = \mathbf{U}^\star$.

Next, Theorem 2.7 guarantees that $\mathbf{z}^\star$ (in which $\mathbf{U}^{1\star} = \cdots = \mathbf{U}^{J\star} = \mathbf{U}^\star$) corresponds to a critical point $(\mathbf{U}^\star, \mathbf{V}^\star)$ of the centralized problem (15), which is exactly equivalent to problem (13). Here, $\mathbf{V}^\star$ is the concatenation of $\mathbf{V}_1^\star, \ldots, \mathbf{V}_J^\star$ as in (14). Theorem 3.1 tells us that problem (13) has two types of critical points: global minimizers and strict saddles. If $(\mathbf{U}^\star, \mathbf{V}^\star)$ were a strict saddle point of (13), Theorem 2.7 tells us that $\mathbf{z}^\star$ must then be a strict saddle of (18). However, $\mathbf{z}^\star$ is almost surely a second-order critical point of (18), where the Hessian has no negative eigenvalues. It follows that $(\mathbf{U}^\star, \mathbf{V}^\star)$ must almost surely be a global minimizer of problem (13).

# 13 Experiments

In our first experiment, we generate a rank-$r$ ground truth matrix $\mathbf{Y} = \begin{bmatrix} \mathbf{Y}_1 & \mathbf{Y}_2 & \cdots & \mathbf{Y}_J \end{bmatrix} \in \mathbb{R}^{n \times Jm_j} \left(\sum_j m_j = m\right)$, where $r = 10$, $n = 50$, $J = 10$, and $m_j = 20$ for all $j$, by multiplying two standard

(a) Distributed matrix completion

(b) Stochastic distributed matrix completion

(c) Distributed matrix sensing

Figure 2: (a) Convergence of DGD+LOCAL for distributed matrix completion. (b) Convergence of stochastic DGD+LOCAL—in each iteration, one node is randomly chosen to perform the update—for distributed matrix completion. (c) Convergence of DGD+LOCAL for distributed matrix sensing.

Gaussian matrices (i.e., each entry is i.i.d. from $\mathcal{N}(0,1)$) of size $n \times r$ and $r \times m$. We solve (15) via (17) with stepsize $\mu = 10^{-3}$. In the left panel of Figure 1, we plot the optimality distance $\sum_{j=1}^{J} \left\| \mathbf{U}^j \mathbf{V}_j^T - \mathbf{Y}_j \right\|_F^2$ and consensus error $\sum_{j=1}^{J} \left\| \mathbf{U}^j - \frac{1}{J} \sum_{i=1}^{J} \mathbf{U}^i \right\|_F^2$ as a function of the number of iterations and verify our theoretical result that for the low-rank matrix factorization problem DGD+LOCAL achieves both global optimality and exact consensus.

In our second experiment, we set $J = 10$ and consider the quadratic least squares optimization problem

$$\underset{\mathbf{x}^1, \mathbf{x}^2, \dots, \mathbf{x}^J}{\text{minimize}} \frac{1}{2} \sum_{j=1}^{J} \left( \mathbf{x}^j - \mathbf{b}_j \right)^T \mathbf{A}_j \left( \mathbf{x}^j - \mathbf{b}_j \right) \tag{25}$$

where $\mathbf{A}_j$ is a $5 \times 5$ randomly generated symmetric matrix with eigenvalues uniformly distributed in $(0,1)$ and $\mathbf{b}_j$ is a $5 \times 1$ standard Gaussian vector. We use standard DGD (because this problem has no local variables $\mathbf{y}_j$) with a stepsize $10^{-2}$ to solve (25) and plot the value of objective function $\frac{1}{2} \sum_{j=1}^{J} \left( \mathbf{x}^j - \mathbf{b}_j \right)^T \mathbf{A}_j \left( \mathbf{x}^j - \mathbf{b}_j \right)$ and consensus error $\sum_{j=1}^{J} \left\| \mathbf{x}^j - \frac{1}{J} \sum_{i=1}^{J} \mathbf{x}^i \right\|_F^2$ in the right panel of Figure 1. We observe convergence only to a neighborhood of the optimal solution with a consensus error proportional to the stepsize.

We also conduct experiments on matrix completion and matrix sensing problems in the distributed setup. For matrix completion, given a rank-$r$ random matrix $\mathbf{Y} \in \mathbb{R}^{n \times m}$ partitioned into $J$ submatrices, i.e., $\mathbf{Y} = \begin{bmatrix} \mathbf{Y}_1 & \mathbf{Y}_2 & \cdots & \mathbf{Y}_J \end{bmatrix}$ with $\mathbf{Y}_j$ of size $n \times m_j$ and $\sum_j m_j = m$, we solve the optimization problem

$$\underset{\mathbf{U}^1, \cdots, \mathbf{U}^J, \mathbf{V}_1, \cdots, \mathbf{V}_J}{\text{minimize}} \sum_{j=1}^{J} f_j \left( \mathbf{U}^j, \mathbf{V}_j \right) \tag{26}$$

where

$$f_j \left( \mathbf{U}^j, \mathbf{V}_j \right) = \frac{1}{2} \sum_{(l,k) \in \Omega_j} \left[ \left( \mathbf{U}^j \mathbf{V}_j^T \right)_{l,k} - (\mathbf{Y}_j)_{l,k} \right]^2$$

and $\Omega_j = \left\{ (l,k) : (\mathbf{Y}_j)_{l,k} \text{ is observed} \right\}$, using DGD+LOCAL with random initialization. In our experiment, we select $n = 50$, $m_j = 5$ for all $j$, $J = 5$, $m = 25$, $r = 2$, and the total number of entries observed in $\mathbf{Y}$ to be $3r(n + m)$.

As shown in Fig 2a, the objective value, recovery error, and consensus error all converge quickly to 0. Similar results are shown in Fig 2b, where we applied a "stochastic" version of DGD+LOCAL (in each iteration one node is randomly chosen to perform the update) to the same matrix completion problem. For matrix sensing, we

use the formulation

$$\underset{\mathbf{U}^1,\cdots,\mathbf{U}^J,\mathbf{V}_1,\cdots,\mathbf{V}_j}{\text{minimize}} \sum_{j=1}^{J} \left\| \mathcal{A}_j \left( \mathbf{U}^j \mathbf{V}_j^T \right) - \mathbf{y}_j \right\|_2^2, \tag{27}$$

where $\mathbf{U}^j \in \mathbb{R}^{n \times r}$, $\mathbf{V}_j \in \mathbb{R}^{m_j \times r}$, $\mathbf{y}_j \in \mathbb{R}^p$ and $\mathcal{A}_j : \mathbb{R}^{n \times m_j} \to \mathbb{R}^p$. For this problem, the sensing mechanism is local to each matrix block:

$$\left( \mathcal{A}_j \left( \mathbf{U}^j \mathbf{V}_j^T \right) \right)_i = \left\langle \mathbf{A}_j^i, \mathbf{U}^j \mathbf{V}_j^T \right\rangle, \ \mathbf{A}_j^i \in \mathbb{R}^{n \times m_j}, \ \forall j = 1, \cdots, J, i = 1, \cdots p.$$

We choose $n = 50$, $m_j = 5$, $J = 5$, $r = 2$, $p = \frac{1}{2}r(n + m_j)$ and use DGD+LOCAL with a random initialization to solve the matrix sensing problem and show the objective value, recovery error, and consensus error in Fig 2c. Again, the objective value, recovery error, and consensus error all converge to 0. Both the matrix completion and matrix sensing problems satisfy the symmetric gradient condition of Proposition 2.3, which explains the convergence to the consensus subspace.