[Reviews · NeurIPS 2019]

Reviewer 1



The authors work on the distributed low-rank matrix factorization problem, while the performance of the proposed method on distributed systems is not validated by experiments. The novelty of this work is not clear. For example, the Proposition 2.1, Definition 2.1. Theorem 2.1, Theorem 2.2 and Theorem 2.3 are copied from existing works. The results in Theorem 2.4 are easy to derive and do not spell out the implication to the main result. In Theorem 2.5, the objective function will converge under the assumption that the sequence ${z(k)}$ is bounded. However, the authors do not prove when this condition will be satisfied. In the literature of MF, there have been a number of works showing that under the incoherence condition, alternating minimization applied to the factors recovers the true matrix. Such kind of statistical guarantee is missing in the work.

Reviewer 2



I've read the response and remain my score. ============================================================== 1. Originality: This paper improved the general distributed gradient descent for matrix factorization under suitable conditions. 2. Quality: The proposed algorithm is simple and easy to implement, and its analysis is comprehensive and beautiful. 3. Clarity: The statements of theorem and its preliminaries are clear. Even though I am not familiar with the technical details in this paper the main idea can be understood. 4. Significance: This paper gives a good point to generalize the standard DGD algorithm.

Reviewer 3



UPDATE AFTER REBUTTAL: Thanks for your responses. I would appreciate if you could incorporate the reposes from the rebuttal into the paper (in case of acceptance). Especially, -better highlighting of the contributions (moving some of the cited auxiliary results to the appendix might also be worth considering) -discussion of stochastic updates -for the promised discussion on alterative approaches: please check again if all primal-dual methods really require a 'star-topology', and not just a connected graph; and include references to the literature; similar with the comment on gossip averaging: the literature discusses already many variants (respective 'orders' of averaging/update steps); so please check again there as well. --- Clarity: Overall, the paper is well written and easy to follow. However, I would encourage to highlight contributions w.r.t. to previous works more clearly (e.g. pages 4-5 consists of a host of theorems cited from previous work, the transition to the new contributions is a bit vague, and also most results look like trivial extensions of these cited results, so commenting on this would help guiding the readers to the novel points). Quality: The theorems all look sound; however, only asymptotic results are provided. Further, comparisons to alternative baselines (instead of DSG) would strengthen the paper. Originality & Significance: As outlined above, the significance could potentially be estimated more favorably if discussion/comparisons to strong baselines could be added. Related work: With many recent advances on the matrix completion problem, such as e.g. also [Matrix Completion has no Spurious Local Minima, https://arxiv.org/pdf/1605.07272.pdf], the results here might not come as a complete surprise (though it is for the decentralized setting). Could the authors comment to connections to this work?

[Author Response · NeurIPS 2019]

We appreciate the elaborate and constructive responses of the reviewers.

**Reviewer 1:**

Q1: *...the proposed method on distributed systems is not validated by experiments*

A1: We initially presented only experiments for distributed matrix factorization

(DMF) in the introduction (Fig. 1 and lines 78–82) with details in Section 13 of

the supplement. We have now also conducted experiments on distributed matrix

completion (DMC) (the same setup as in DMF, except that only $3r \max\{m, n\}$

entries of the rank-$r$ matrix $\boldsymbol{Y} \in \mathbb{R}^{n \times m}$ are observed) using DGD+LOCAL with

random initialization. As shown at right, the objective value, recovery error, and

consensus error all converge quickly to $0$. Similar results are shown for distributed

matrix sensing (DMS). We will incorporate these experiments into the final paper.

Distributed Matrix Completion

Distributed Matrix Sensing

Q2: *Novelty not clear... connection to existing works...*

A2: We agree that many of the results in Section 2.2.1 are from existing works as we cited, but we note that these

are not our main results. Since the objective function in DMF does not satisfy the common assumption of a globally

Lipschitz gradient, Theorem 2.4 extends Theorem 2.3 for functions with a locally Lipschitz gradient.

That being said, we realize that we did not communicate our work with adequate precision. Our **first main contribution**

comprises the algorithmic and geometric results for DGD+LOCAL: $(i)$ Section 2.2.2 shows that DGD+LOCAL will

converge to a second-order critical point of the regularized objective function (7), and $(ii)$ Section 2.3 provides

conditions under which the geometric landscape of the distributed objective function (7) is "equivalent" to the geometric

landscape of the original centralized objective function, ensuring *exact consensus* of DGD+LOCAL, in contrast to

general DGD results which admit consensus error proportional to the stepsize [15,23]. Our **second main contribution**

is the result in Section 3 showing consensus and global optimality of DGD+LOCAL for DMF. In the revision, we will

highlight the novelty and contributions w.r.t. previous works more clearly in the introduction.

Q3: *Theorem 2.5,... under the assumption that $\{z(k)\}$ is bounded... do not prove when this condition will be satisfied.*

A3: By assuming only Lipschitz gradient and the KL inequality, in general one cannot guarantee boundedness. Thus, it

is commonly assumed that the sequence is bounded, e.g., in [1, 2] and Theorems 2.1–2.2. If we further assume that the

function is coercive, then the generated sequence is bounded. We will incorporate this discussion in the revision.

Q4: *In the literature of MF, there have been a number of works... such kind of statistical guarantee is missing.*

A4: We think the reviewer may be referring to statistical guarantees for matrix completion. This paper focuses on MF,

but the experiment in Q1 demonstrates the potential to extend DGD+LOCAL results to DMC. In this case, we believe

the existing statistical guarantees for DMC can be directly applied thanks to the equivalence of the geometric landscape

between the centralized and distributed objective functions. This is the subject of future work.

**Reviewer 2:** We appreciate the positive comments and will polish the presentation of the theorems.

**Reviewer 3:**

Q1: *Overall, the paper is well written and easy to follow. However, I would encourage to highlight contributions ...*

A1: This is a great suggestion and we will highlight our results in a more transparent way, as also requested by R1.

Q2: *The theorems all look sound; however, only asymptotic results are provided.*

A2: The (asymptotic) convergence rate (which is at least sublinear depending on the KL exponent) of DGD+LOCAL

can be obtained by using the KL framework in [1,2], as used in [Proposition 2, 24]. Surprisingly, both Fig. 1 and the top

right figure suggest that DGD+LOCAL for DMF converges at a *linear* rate. We will incorporate this discussion and

leave the investigation to future work.

Q3: *A discussion of such alternative approaches ... and comparison would clearly strengthen the paper...*

A3: Compared with primal-dual methods which require a "star topology", DGD+LOCAL (or DGD) can be applied to

ring networks without a central node. DGD+LOCAL is similar in spirit to gossip-based methods, but differs in the

order of performing the local update and local averaging steps. This small difference allows us to view the proposed

algorithm as performing GD on a regularized objective function and thus prove convergence to a global minimum by

geometric analysis of that function. We will incorporate this discussion.

Q4: *allowing for stochastic gradient updates could increase the impact*

A4: This is a great suggestion. Note that the geometric analysis in Section 2.3 is for

the objective function and can be utilized to guarantee convergence of any iterative

algorithm. We apply "Stochastic DGD+LOCAL" (in each iteration we randomly

chose one node to update) to DMC and show the result at right. We will incorporate

this and similar discussion for DMS.

Stochastic Distributed Matrix Completion

Q5: *Could the authors comment to connections to the work [Matrix Completion has no Spurious Local Minima]?*

A5: We apologize for the omission. Similar to [8,18], that work provides geometric analysis of the *centralized* MC

problem, while part of our work focuses on the geometric analysis of the distributed problem. By connecting these two

formulations, we show the distributed problem must inherit the same benign geometry.

[Meta-Review · NeurIPS 2019]

The paper studies distributed matrix factorization problems. It takes the view that distributed (sub)gradient descent relates to a regularized version of the original optimization problem, and then shows that stationary points of the distributed matrix completion problem satisfy the consensus constraint. While the significance of the paper caused some discussions, the reviewers remained mostly positive in the final assessment, resulting in a narrow accept decision. We hope the suggested changes and comments help improve the paper for the camera ready version, in particular, this concerns the clarity of contributions, related work on primal-dual and gossip methods as mentioned e.g. by Reviewer 3 and other comments by the reviewers.